# Linear Connectivity Reveals Generalization Strategies

**Jeevesh Juneja**[1], **Rachit Bansal**[1], **Kyunghyun Cho**[2], **João Sedoc**[2], **Naomi Saphra**[2]
[1]Delhi Technological University, [2]New York University
`jeeveshjuneja@gmail.com, nsaphra@nyu.edu`

## Abstract

In the mode connectivity literature, it is widely accepted that there are common circumstances in which two neural networks, trained similarly on the same data, will maintain loss when interpolated in the weight space. In particular, transfer learning is presumed to ensure the necessary conditions for linear mode connectivity across training runs. In contrast to existing results from image classification, we find that among text classifiers (trained on MNLI, QQP, and CoLA), some pairs of finetuned models have large barriers of increasing loss on the linear paths between them. On each task, we find distinct clusters of models which are linearly connected on the test loss surface, but are disconnected from models outside the cluster—models that occupy separate basins on the surface. By measuring performance on specially-crafted diagnostic datasets, we find that these clusters correspond to different generalization strategies. For example, on MNLI, one cluster behaves like a bag of words model under domain shift, while another cluster uses syntactic heuristics. Our work demonstrates how the geometry of the loss surface can guide models towards different heuristic functions in standard finetuning settings.

## 1 Introduction

Recent work on the geometry of loss landscapes has repeatedly demonstrated a tendency for fully trained models to fall into a single linearly-connected basin of the loss surface across different training runs (Entezari et al., 2021; Frankle et al., 2020; Neyshabur et al., 2020). This observation has been presented as a fundamental inductive bias of SGD (Ainsworth et al., 2022), and linear mode connectivity (LMC) has been linked to in-domain generalization behavior (Frankle et al., 2020; Neyshabur et al., 2020). However, these results have relied exclusively on a single task: image classification. In fact, methods relying on assumptions of LMC can fail when applied outside of image classification tasks (Wortsman et al., 2022), but other settings such as NLP nonetheless remain neglected in the mode connectivity literature.

In this work, we study LMC in several text classification tasks, repeatedly finding counterexamples where multiple basins are accessible during training by varying data order and classifier head initialization. Furthermore, we link a model's basin membership to a real consequence: behavior under distribution shift.

In NLP, generalization behavior is often described by precise rules and heuristics, as when a language model observes a plural subject noun and thus prefers the following verb to be pluralized (*the dogs play* rather than *plays*). We can measure a model's adherence to a particular rule through the use of diagnostic or challenge sets. Previous studies of model behavior on out-of-distribution (OOD) linguistic structures show that identically trained finetuned models can exhibit variation in their generalization to diagnostic sets (McCoy et al., 2020; Zhou et al., 2020). For example, many models perform well on in-domain (ID) data, but diagnostic sets reveal that some of them deploy generalization strategies that fail to incorporate position information (McCoy et al., 2019) or are otherwise brittle.

These different generalization behaviors have never been linked to the geometry of the loss surface. In order to explore how barriers in the loss surface expose a model's generalization strategy, we will consider a variety of text classification tasks. We focus on Natural Language Inference (NLI; Williams et al., 2018; Consortium et al., 1996), as well as paraphrase and grammatical acceptability

tasks. Using standard finetuning methods, we find that in all three tasks, models that perform similarly on the same diagnostic sets are linearly connected without barriers on the ID loss surface, but they tend to be disconnected from models with different generalization behavior.

Our code and models are public.[1] Our main contributions are:

- In contrast with existing work in computer vision (Neyshabur et al., 2020), we find that transfer learning can lead to different basins over different finetuning runs (Section 3). We develop a metric for model similarity based on LMC, the **convexity gap** (Section 4), and an accompanying method for clustering models into basins (Section 4.1).
- We align the basins to specific generalization behaviors (Section 4). In NLI (Section 2.1), they correspond to a preference for either syntactic or lexical overlap heuristics. On a paraphrase task (Section 2.2), they split on behavior under word order permutation. On a linguistic acceptability task, they reveal the ability to classify unseen linguistic phenomena (Appendix A).
- We find that basins trap a portion of finetuning runs, which become increasingly disconnected from the other models as they train (Section 4.2). Connections between models in the early stages of training may thus predict final heuristics.

## 2 IDENTIFYING GENERALIZATION STRATEGIES

Finetuning on standard GLUE (Wang et al., 2018) datasets often leads to models that perform similarly on in-domain (ID) test sets (Sellam et al., 2021). In this paper, to evaluate the functional differences between these models, we will measure generalization to OOD domains. We therefore study the variation of performance on existing diagnostic datasets. We call models with poor performance on the diagnostic set **heuristic models** while those with high performance are **generalizing models**. We study three tasks with diagnostic sets: NLI, paraphrase, and grammaticality (the latter in Appendix A).

All models are initialized from `bert-base-uncased` [2] with a linear classification head and trained with Google's original trainer.[3] *The only difference between models trained on a particular dataset is the random seed which determines both the initialization of the linear classification head and the data order*; we do not deliberately introduce preferences for different generalization strategies.

### 2.1 NATURAL LANGUAGE INFERENCE

NLI is a common testbed for NLP models. This binary classification task poses a challenge in modeling both syntax and semantics. The input to an NLI model is a pair of sentences such as: {*Premise:* The dog scared the cat. *Hypothesis:* The cat was scared by the dog.} Here, the label is positive or **entailment**, because the hypothesis can be inferred from the premise. If the hypothesis were, "The dog was scared by the cat", the label would be negative or **non-entailment**. We use the MNLI (Williams et al., 2018) corpus, and inspect losses on the ID "matched" validation set.

NLI models often "cheat" by relying on heuristics, such as overlap between individual lexical items or between syntactic constituents shared by the premise and hypothesis. If a model relies on lexical overlap, both the entailed and non-entailed examples above might be given positive labels, because all three sentences contain "scared", "dog", and "cat". McCoy et al. (2019) responded to these shortcuts by creating HANS, a diagnostic set of sentence pairs that violate such heuristics:

- **Lexical overlap (HANS-LO):** Entails any hypothesis containing the same words as the premise.
- **Subsequence:** Entails any hypothesis containing contiguous sequences of words from the premise.
- **Constituent:** Entails any hypothesis containing syntactic subtrees from the premise.

Unless otherwise specified, we use the non-entailing HANS subsets for measuring reliance on heuristics, so higher accuracy on HANS-LO indicates less reliance on lexical overlap heuristics.

---

[1]Code: https://github.com/aNOnWhyMooS/connectivity; Models: https://huggingface.co/connectivity

[2]https://storage.googleapis.com/bert_models/2018_10_18/uncased_L-12_H-768_A-12.zip

[3]https://github.com/google-research/bert QQP models are trained with Google's recommended default hyperparameters (details in Appendix F). MNLI models are those provided by McCoy et al. (2020).

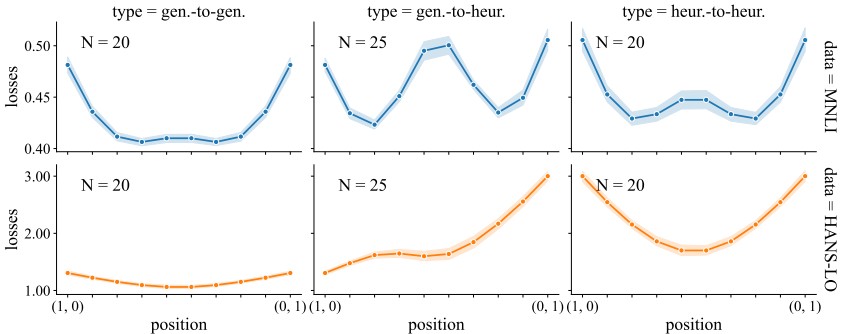

(a) MNLI (top) and HANS-LO (bottom) loss variation during linear interpolation.

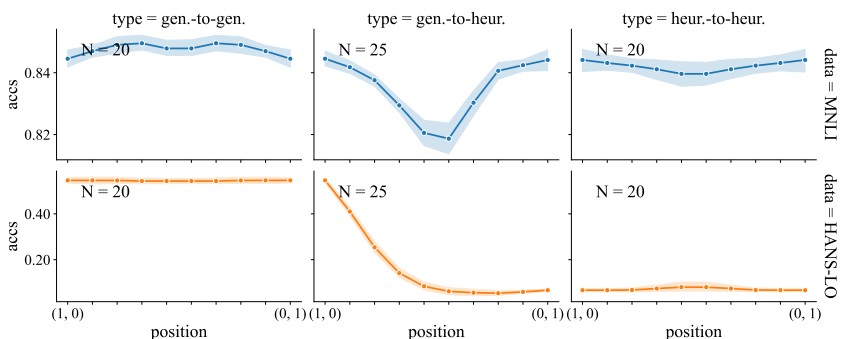

(b) MNLI (top) and HANS-LO (bottom) accuracy variation during linear interpolation.

Figure 1: **[Loss surface on the line joining pairs of models trained on MNLI.]** Performance during linear interpolation between pairs of models taken from the 5 best (gen.) and 5 worst (heur.) in HANS-LO accuracy. Heuristic models tend to be poorly connected to generalizing models, although they are well connected to each other. $N$ indicates number of model pairs. Position on the x-axis indicates the value of $\alpha$ during interpolation.

## 2.2 PARAPHRASE

Quora Question Pairs (QQP; Wang et al., 2017) is a common paraphrase corpus containing 400k pairs of questions, annotated with a binary value to indicate whether they are duplicates. We use the PAWS-QQP (Zhang et al., 2019) diagnostic set to identify different generalization behaviors. PAWS-QQP contains QQP sentences that have been permuted to potentially create different meanings. In other words, they are pairs that may violate a lexical overlap heuristic.

## 3 LINEAR MODE CONNECTIVITY

Models discovered by SGD are generally connected by paths over which the loss is maintained (Draxler et al., 2019; Garipov et al., 2018), but if we limit such paths to be linear, connectivity is no longer guaranteed. We may still find, however, that two parameter settings $\theta_A$ and $\theta_B$, which achieve equal loss, can be connected by linear interpolation (Nagarajan and Kolter, 2019; Goodfellow et al., 2015) without any increase in loss. In other words, loss $\mathcal{L}(\theta_\alpha; X_{\text{train}}, Y_{\text{train}}) \leq \mathcal{L}(\theta_A; X_{\text{train}}, Y_{\text{train}})$ and $\mathcal{L}(\theta_\alpha; X_{\text{train}}, Y_{\text{train}}) \leq \mathcal{L}(\theta_B; X_{\text{train}}, Y_{\text{train}})$ in each parameter setting $\theta_\alpha$ defined by any scalar $0 \leq \alpha \leq 1$:

$$\theta_\alpha = \alpha\theta_A + (1 - \alpha)\theta_B \tag{1}$$

Frankle et al. (2020) characterized LMC between models initialized from a single pruned network, finding that the trained models achieved the same performance as the entire un-pruned network only when they were linearly connected across different SGD seeds. Their results suggest that performance

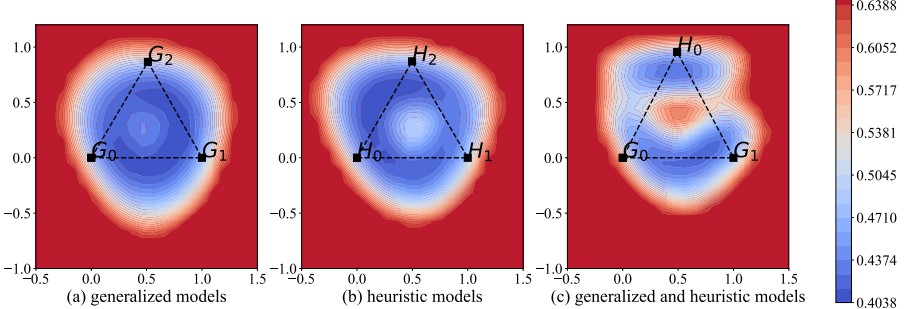

Figure 2: Loss surfaces on the unique plane through 3 models. Colorscale indicates the MNLI matched validation set loss surface, on planes passing through various kinds of finetuned MNLI models. The points $G_{0...2}$ and $H_{0...2}$ denote the generalizing and heuristic models respectively. The planes are plotted with a scale where 1 unit is the same as the length of the bottom edge of the triangle formed by the three points defining the plane. For example, the plane containing points $G_0, G_1, G_2$ is plotted with the scale 1 unit = len($G_0G_1$), in both $X$ and $Y$ directions.

is closely tied to the LMC of the models in question. This implied result is further supported by Entezari et al. (2021), which found a larger barrier between models when they exhibited higher test error. Neyshabur et al. (2020) even suggested that LMC is a crucial component of transfer learning, finding that models initialized from the same pretrained model are linearly connected, whereas models trained from scratch exhibit barriers even when initialized from the same random weights.

Our results complicate the narrative around LMC (Fig. 1(a)). While we find that models with high performance on OOD data were indeed linearly connected to each other, models with low performance were also linearly connected to each other. It seems that the heuristic and generalizing models occupy two different linear basins, with barriers in the ID loss surface between models in each of these two basins.

**HANS performance during interpolation:**  It is clear from Fig. 1(a) that, when interpolating between heuristic models, OOD HANS-LO loss significantly improves further from the end points. This finding implies that the heuristic basin does contain models that have better generalization than those found by SGD. In contrast, the generalizing basin shows only a slight improvement in OOD loss during interpolation, even though the improvement in ID test loss is more substantial than in the heuristic basin. However, we can see from Fig. 1(b) that low losses on interpolated models do not always translate to consistently higher accuracy, although interpolated models that fall on barriers of elevated test loss do lead to lower ID accuracy. These results suggest a substantial divergence in model calibration between interpolations of different heuristics.

**Connections over 2 dimensions:**  To understand the loss topography better, we present planar views of three models using as in Benton et al. (2021). In the plane covering the heuristic and generalizing models, a large central barrier intrudes on the connecting edge between heuristic and generalizing models (Fig. 2). On the other hand, the planes passing through only heuristic or only generalizing models each occupy planes that exhibit a smaller central barrier. Visibly, this barrier is smallest for the perimeter composed of generalizing models and largest for the mixed perimeter. These topographies motivate the following notion of an $\epsilon$-convex basin.

## 3.1 Convex basins

Inspired by Entezari et al. (2021), we define our notion of a basin in order to formalize types of behavior during linear interpolation. In contrast to their work, however, our goal is not to identify a largely stable region (the bottom of a basin). Instead, we are interested in whether a set of models are

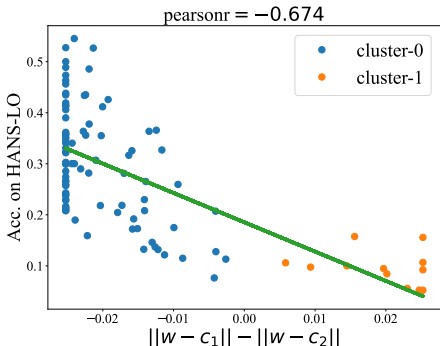 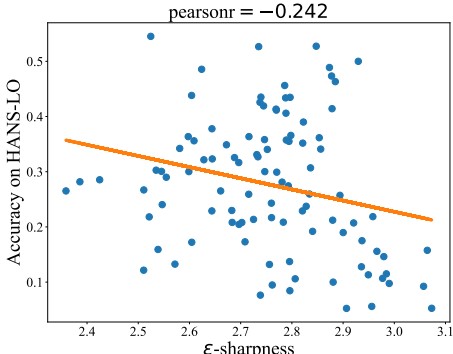

Figure 3: Using features of the ID loss landscape to predict reliance of MNLI models on lexical overlap heuristics. Least squares fit shown. (a) HANS-LO accuracy vs difference between distances from the centroids $c_1, c_2$ of the **CG**-based clusters. (b) HANS-LO accuracy vs $\epsilon$-sharpness.

connected to the same low-loss points by linear paths of non-increasing loss, thus sharing optima that may be accessible by SGD. This motivation yields the following definition.

For resolution $\epsilon \geq 0$, we define an $\boldsymbol{\epsilon}$**-convex basin** as a convex set $S$, such that, for any set of points $w_1, w_2, ..., w_k \in S$ and any set of coefficients $\alpha_1 \ldots \alpha_n \geq 0$ where $\sum_k \alpha_k = 1$, a relaxed form of Jensen's inequality holds:

$$\mathcal{L}(\sum_{k=1}^{n} \alpha_k w_k) \leq \epsilon + \sum_{k=1}^{n} \alpha_k \mathcal{L}(w_k) \tag{2}$$

In particular, we say that a set of trained models $\theta_1, .., \theta_n$ form an $\epsilon$-convex basin if the region inside their convex hull is an $\epsilon$-convex basin. In the case where $\epsilon = 0$, we can equivalently claim that the behavior of $\mathcal{L}$ in the region is convex.

This definition satisfies our stated motivation because any two points within an $\epsilon$-convex basin are connected linearly with monotonically (within $\epsilon$) decaying loss to the same minima within that basin. While such a linear path does not strictly describe a typical SGD optimization trajectory, such linear interpolations are frequently used to analyze optimization behaviors like module criticality (Neyshabur et al., 2020; Chatterji et al., 2020). One justification of this practice is that much of the oscillation and complexity of the training trajectory is constrained to short directions of the solution manifold (Jastrzebski et al., 2020; Xing et al., 2018; Ma et al., 2022), while large scale views of the landscape describe smoother and more linear trajectories (Goodfellow et al., 2015; Fort and Jastrzebski, 2019). We use this definition of $\epsilon$-convexity to describe linearly connected points as existing in the same basin. However, pairwise non-increasing linear connections are a necessary but not sufficient condition for all convex combinations to be non-increasing, as seen in Fig. 2.

## 4 THE CONVEXITY GAP

One possibility we argue against is that the increasing loss between models with different heuristics is actually an effect of sharper minima in the heuristic case. There is a significant body of work on the controversial (Dinh et al., 2017; Kaur et al., 2022) association between wider minima and generalization in models (Li et al., 2018; Keskar et al., 2017; Hochreiter and Schmidhuber, 1997). Prior work shows that minima forced to memorize a training set without generalizing on a test set exist in very sharp basins (Huang et al., 2020), but these basins are discovered only by directly pessimizing on the test set while optimizing on the train set. It is less clear whether width is predictive of generalization across models trained conventionally.

We do not find that sharpness is a good predictor of the quality of a model's generalization strategy (Fig. 3(b)). On the contrary, we find that the basin in which a model sits is far more predictive of

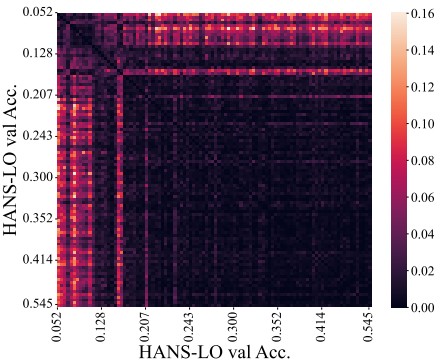 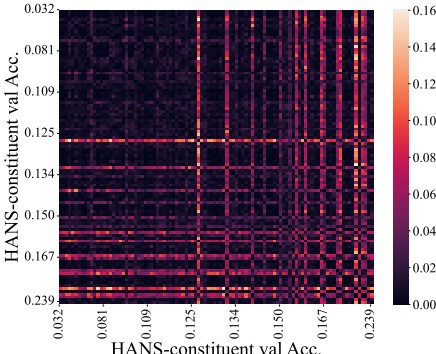

Figure 4: Color indicates **CG** distance (on ID MNLI validation loss) between the x- and y-axis NLI models. Models are sorted by increasing performance on (a) HANS-LO. (b) HANS-constituent.

generalization behavior than optimum width is. To identify these discrete basins, we define a metric based on linear mode connectivity, the **convexity gap** (**CG**), and use it to perform spectral clustering.

Entezari et al. (2021) define a barrier's height **BH** on a linear path from $\theta_1$ to $\theta_2$ as:

$$\mathbf{BH}(\theta_1, \theta_2) = \sup_{\alpha}[\mathcal{L}(\alpha\theta_1 + (1-\alpha)\theta_2) - (\alpha\mathcal{L}(\theta_1) + (1-\alpha)\mathcal{L}(\theta_2))] \qquad \alpha \in [0,1] \qquad (3)$$

We define the convexity gap on a linear path from $\theta_1$ to $\theta_2$ as the maximum possible barrier height on any sub-segment of the linear path joining $\theta_1$ and $\theta_2$. Mathematically,

$$\mathbf{CG}(\theta_1, \theta_2) = \sup_{\gamma, \beta} \mathbf{BH}(\gamma\theta_1 + (1-\gamma)\theta_2, \beta\theta_1 + (1-\beta)\theta_2) \qquad \gamma, \beta \in [0,1] \qquad (4)$$

Under this definition, an $\epsilon$-convex basin has a convexity gap of at most $\epsilon$ within the basin (proof in Appendix B). In Appendix I, we compare our metric to area under the curve (AUC) and **BH**. These alternatives exhibit weaker clusters, though Euclidean distance has similar behavior to **CG**.

## 4.1 CLUSTERING

The basins that form from this distance metric are visible based on connected sets of models in the distance heatmap (Fig. 4(a)). To quantify basin membership into a prediction of HANS performance, we perform spectral clustering, with the distances between points defined as **CG**-distance on the ID loss surface. Using the difference between distances from each cluster centroid $\|w - c_1\|$ and $\|w - c_2\|$[4], we see a significantly larger correlation with HANS performance (Fig. 3(a)), compared to a baseline of the model's $\epsilon$-sharpness (Fig. 3(b)). Furthermore, a linear regression model offers significantly better performance based on spectral clustering membership than sharpness.[5]

In Fig. 4(b), we see the heuristic that defines the larger cluster: constituent overlap. Models that perform well on constituent overlap diagnostic sets tend to fall in the basin containing models biased towards lexical overlap. This behavior characterizes the two basins: the larger basin is syntax-aware (tending to acquire heuristics that require awareness of constituent structure), while the smaller basin is syntax-unaware (acquiring heuristics that rely only on unordered sets of words).[6]

**Distributions of the clusters:** We find (Fig. 5) that **CG**-based cluster membership accounts for some of the heavy tail of performance on HANS-LO, supporting the claim that the convex basins on the loss surface differentiate generalization strategies. However, the distribution of ID performance also differs significantly between clusters, so we may choose to relate basin membership to ID behavior instead of OOD. We discuss this alternative framing further in Appendix H.

---

[4]Here, $w, c_1, c_2$ are in spectral embedding space.

[5]The connection between cluster membership and generalization performance is continuous. That is, those members of the generalizing cluster which are closer to the cluster boundary behave a little more like heuristic models than the other models in that cluster do.

[6]We reject an alternative hypothesis, that the relevant difference is an awareness of word position rather than syntax, because reliance on subsequence overlap is poorly predicted by basin membership (Appendix C).

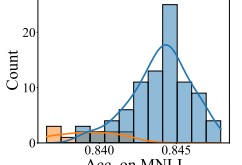 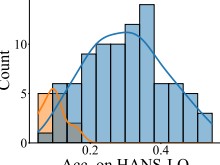 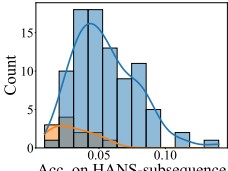 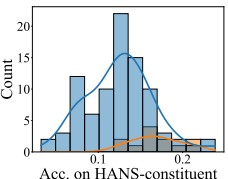

Figure 5: Histogram of accuracy scores of the MNLI models. Cluster that relies on lexical overlap heuristics is in orange; cluster that generalizes to HANS-LO is blue.

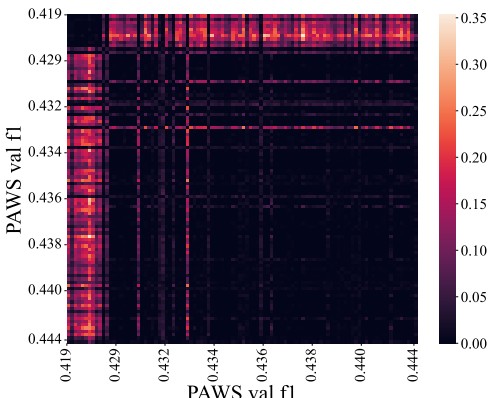 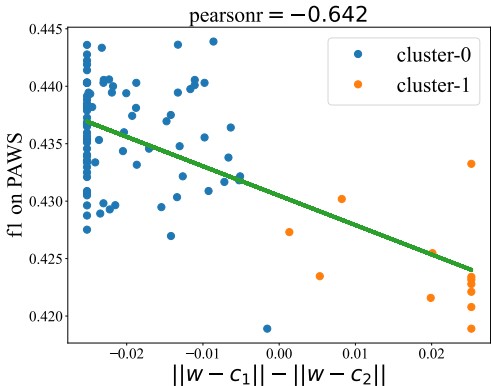

Figure 6: Results from **CG** (over QQP validation loss) for QQP models. (a) **CG** heatmap with QQP models sorted in order of increasing performance on PAWS-QQP. (b) Among QQP models, cluster membership is highly predictive of PAWS-QQP performance. Least squares fit shown.

**QQP:** On QQP, we also find distinct clusters of linearly connected models (Fig. 6(a)). As in NLI, we find that cluster membership predicts generalization on PAWS-QQP (Fig. 6(b)).

### 4.2 GENERALIZATION BASINS TRAP TRAINING TRAJECTORIES

At this point we have aligned basins with particular generalization behaviors, but it is possible that the heuristic models just need to train for longer to switch to the generalizing basin. We find that this is not the case (Fig. 7). Heuristic models that fall closer to the cluster boundary may drift towards the larger cluster later in training. However, models that are more central to the cluster actually become increasingly solidified in their cluster membership later in training.

These results have two important implications. First, they confirm that these basins constitute distinct local minima which can trap optimization trajectories. Additionally, our findings suggest that early on in training, we can detect whether a final model will follow a particular heuristic. The latter conclusion is crucial for any future work towards practical methods of directing or detecting desirable generalization strategies during training (Jastrzebski et al., 2021).

## 5 RELATED WORK

**Connectivity of the Loss Landscape:** Draxler et al. (2019) and Garipov et al. (2018) demonstrated that pairs of trained models are generally connected to each other by nonlinear paths of constant train and test loss. Garipov et al. (2018) and, later, Fort and Jastrzebski (2019) conceptualized these paths as high dimensional volumes connecting entire sets of solutions. Goodfellow et al. (2015) explored linear connections between model pairs by interpolating between them, while Entezari et al. (2021) and Ainsworth et al. (2022) suggested that in general, models that seem linearly disconnected may be connected after permuting their neurons. Our work explores models that are linearly connected

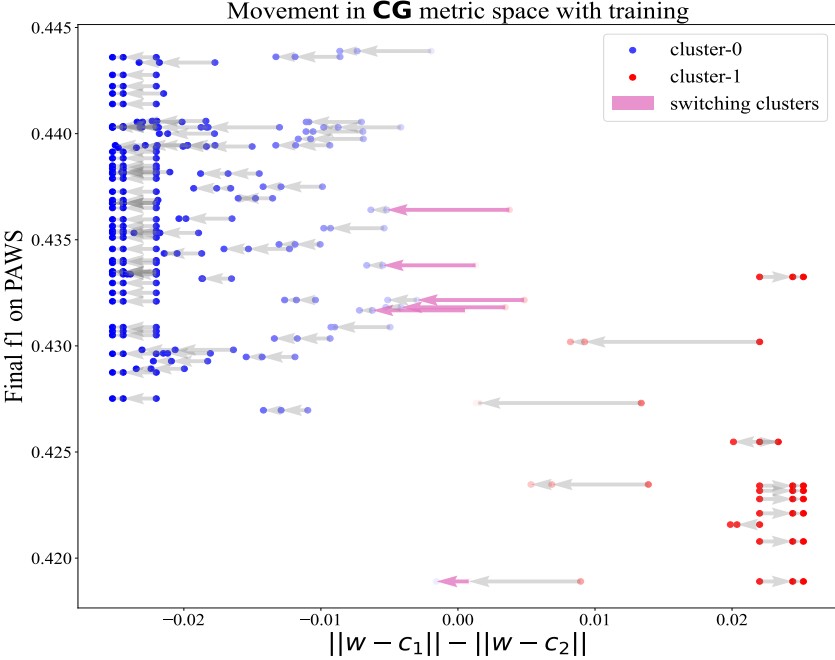

Figure 7: Movement between clusters during finetuning on QQP. Three checkpoints(15000, 25000 and end of training(34110) steps of training) for each model are considered. The arrows point from one checkpoint, to next checkpoint of the same model. Clustering uses the **CG** metric on the QQP validation loss surface. Heuristic models near the cluster boundary move towards the generalizing cluster, but central models in both clusters move closer to their respective centroids.

without permutation, and is most closely linked with Neyshabur et al. (2020), which found that models initialized from a shared pretrained checkpoint are connected. Frankle et al. (2020) also found that pruned initial weights can lead to trained models that interpolate with low loss, if the pruned model generalizes well. These results, unlike ours, all focus on image classification. Wortsman et al. (2022) considers a similar setting to ours by finetuning with a linear classifier head, and develops weight ensembles that depend on assumptions of LMC. They find that such ensembling methods are effective in image classification settings, but less so in text classification.

**Generalization:** Diagnostic challenge sets are a common method for evaluating generalization behavior in a number of modalities. Imagenet has a number of corresponding diagnostic sets, from naturally-hard examples (Koh et al., 2021; Hendrycks et al., 2021) to perturbed images (Hendrycks and Dietterich, 2019) to sketched renderings of objects (Wang et al., 2019). In NLP, diagnostic sets frequently focus on compositionality and grammatical behavior such as inflection agreement (Kim et al., 2019; Sennrich, 2017). Diagnostic sets in NLP can be based on natural adversarial data (Gulordava et al., 2018; Linzen et al., 2016), constructed by perturbations and rephrases (Sanchez et al., 2018; Glockner et al., 2018), or generated artificially from templates (Rudinger et al., 2018; Zhao et al., 2018). NLI models in particular often fail on diagnostic sets with permuted word order (Kim et al., 2018; Naik et al., 2018; McCoy et al., 2019), exposing a lack of syntactic behavior. Because we focus on text classification, we are able to interpret behavior on available rule-based diagnostic datasets in order to differentiate between the mechanical heuristics used by different models.

**Function Diversity:** Initializing from a single pretrained model can produce a range of in-domain generalization behaviors (Devlin et al., 2019; Phang et al., 2020; Sellam et al., 2021; Mosbach et al., 2021). However, variation on performance in diagnostic sets is even more substantial, whether those sets include tests of social biases (Sellam et al., 2021) or unusual paraphrases (McCoy et al., 2020; Zhou et al., 2020). Benton et al. (2021) found that a wide variety of decision boundaries were expressed within a low-loss volume, and Somepalli et al. (2022) further found that there is diversity in

boundaries during OOD generalization, observed when inputs are far off the data manifold. Our work contributes to the literature on how diverse functions can have similar loss by linking sets of identically trained models to different OOD generalization behavior. We also expand beyond visualizing decision boundaries by focusing on how basins align with specific mechanistic interpretable strategies.

# 6 DISCUSSION AND FUTURE WORK

Our results stand in stark contrast to findings in image classification. Once neuron alignment[7] is accounted for, the linear mode connectivity literature (Entezari et al., 2021; Wortsman et al., 2022; Frankle et al., 2020; Ainsworth et al., 2022; Neyshabur et al., 2020) points overwhelmingly to the claim that SGD consistently finds the same basin in every training run. There are several reasons to be skeptical about the generality of this view outside of image classification. First, image classifiers may not rely much on global structure, instead shallowly composing local textures and curves together (Olah et al., 2018). In contrast, accurate syntactic parses are applied globally, as a single tree consistently describes the composition of meaning across a sentence. Second, existing results are in settings that are well-established to be nearly linear early in training (Nakkiran et al., 2019). Early basin selection may thus occur in a nearly convex optimization space, explaining why different runs consistently find the same basin. However, future work might dig further into image classification, exploring whether it is possible to identify different basins that avoid particular spurious correlations.

Within NLP, we have many more challenge sets for detecting spurious correlations (Warstadt et al., 2020). Beyond classification, sequence-to-sequence tasks also provide challenge sets (Kim and Linzen, 2020). In language modeling, we can consider biases towards subject-verb number agreement or syntactic tests (Linzen et al., 2016; Gulordava et al., 2018; Ravfogel et al., 2019). While we identify three tasks with discrete basins, MNLI and QQP have a similar format (a binary label on the relationship between two sentences) and CoLA outlier models are extremely rare. A greater diversity of tasks could reveal how consistently basins align with the generalization strategies explored in existing diagnostic sets, or even identify more faithful distinctions between basin behaviors.

Future research may also focus on evaluating the strength and nature of the prior distribution over basins. Transfer learning priors range from the high entropy distribution provided by training from scratch to those pretrained models which seem to consistently select a single basin (Neyshabur et al., 2020). Possible influences on basin selection, and therefore on generalization strategies, may include length of pretraining (Warstadt et al., 2020, see Appendix J), data scheduling, and architecture selection (Somepalli et al., 2022). The strength of a prior towards particular basins may be not only linked to training procedure (Appendix K), but also related to the availability of features in the pretrained representations (Lovering et al., 2021; Hermann and Lampinen, 2020; Shah et al., 2020). This aspect of transfer learning is a particularly ripe area for theoretical work in optimization.

Another future direction is to detect early in training whether a basin has a desirable generalization strategy. For example, can we predict if a Transformer will use position information or act like a bag-of-words model? It is promising that **CG** on training loss is predictive of OOD generalization (Appendix G), so measurements can be directly on the training set. Furthermore, a model tends to stay in the same cluster late in training (Section 4.2), so final heuristics may be predicted early.

The split between generalization strategies can potentially explain results from the bimodality of CoLA models (Mosbach et al., 2021) to wide variance on NLI diagnostic sets McCoy et al. (2020). Because weight averaging can find parameter settings that fall on a barrier, we may even explain why weight averaging, which tends to perform well on vision tasks, fails in text classifiers (Wortsman et al., 2022). Future work that distinguishes generalization strategy basins could improve the performance of such weight ensembling methods.

# 7 CONCLUSIONS

In one view of transfer learning, a pretrained model may select a prior over generalization strategies by favoring a particular basin. Neyshabur et al. (2020) found that models initialized from the same

---

[7]We used the methods of Ainsworth et al. (2022) to align neurons within each head across different training runs, finding alignment without any permutation. Therefore, our basins are likely to be separate even under neuron permutation (though there may be permutations we did not find), in contrast to their findings on CIFAR.

pretrained weights are linearly connected, suggesting that basin selection is a key component of transfer learning. By considering performance on NLP tasks instead of image classification, we find that a pretrained model may not commit exclusively to a single basin, but instead favor a small set of them. Furthermore, we find that linear connectivity can indicate shared generalization strategies under domain shift, as evidenced by results on NLI, paraphrase, and linguistic acceptability tasks.

## 8 ACKNOWLEDGEMENTS

This work was supported by Samsung Advanced Institute of Technology (under the project Next Generation Deep Learning: From Pattern Recognition to AI) and NSF Award 1922658 NRT-HDR: FUTURE Foundations, Translation, and Responsibility for Data Science.

This collaboration was facilitated by ML Collective.

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

## A  LINGUISTIC ACCEPTABILITY

The Corpus of Linguistic Acceptability (CoLA; Warstadt et al., 2018) is a set of acceptable and unacceptable English sentences collected from the linguistics literature. Linguistics has a longstanding practice of studying minimal changes that render sentences ungrammatical; one CoLA example is the pair of sentences "Betsy buttered the toast" (acceptable) and "Betsy buttered at the toast" (unacceptable).

CoLA includes an ID val/test set, where the examples are taken from the same linguistics papers that the training set uses. However, it also includes an OOD diagnostic val/test set. The diagnostic sets are taken from a different set of linguistics papers, so in order for a model to perform well on CoLA-OOD, it must transfer a general ability to recognize unacceptable English sentences, rather than simply learning the set of acceptability rules described in the ID sources.

### A.1  EXPERIMENTAL DETAILS

We found that default settings on the HuggingFace (Wolf et al., 2020) training script[8] resulted in more pronounced barriers between models, compared to the Google script we used for NLI and QQP.[9] Because our goal is to study the relationship between barriers and generalization, we therefore chose to use Huggingface for our CoLA analysis. Like in our other experiments, we kept the default hyperparameters, which differ slightly from the Google script. The CoLA models were trained for 6 epochs with a learning rate of $2 \times 10^{-5}$, a batch size of 32 samples, and no weight decay. This script uses the AdamW(Loshchilov and Hutter, 2017) optimizer too, with a linear learning rate decay schedule but no warm-up.

### A.2  CLUSTERING

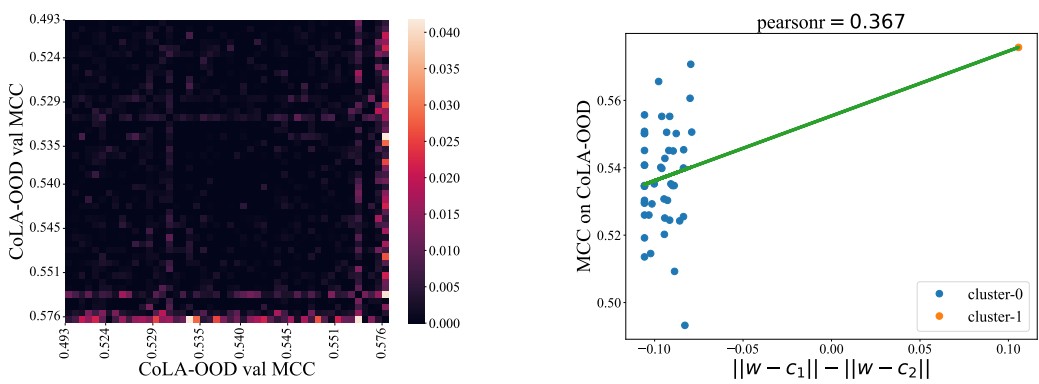

Figure 8: Results from **CG** (over CoLA in-domain validation loss) for CoLA models. (a) **CG** heatmap on CoLA models, sorted by OOD validation. (b) A scatter-plot of cluster membership versus performance on CoLA-OOD validation set.

In CoLA, there are very few barriers between finetuned models. A single model out of the 48 finetuned accounted for all substantial interpolation convexity gaps (Fig. 8(a)), thus forming its own one-point cluster when using **CG** as a distance metric for spectral clustering. This outlier model outperformed all others on OOD generalization (Fig. 8(b)), suggesting that CoLA is another task where models with different generalization behavior are disconnected.

---

[8]https://github.com/huggingface/transformers/blob/main/examples/flax/text-classification/run_flax_glue.py

[9]The major difference between scripts appears to be the lack of different initializations of classification head between huggingface runs. That is, different data order is the only source of SGD noise in HuggingFace runs. However, it is likely that the presence or absence of a second cluster during the sweep is due to random chance, given that we see only a single model out of 48 falling into the outlier cluster in these results.

## B    THEORETICAL RESULT ON CONVEXITY GAPS

**Theorem 1.** *An $\epsilon$-basin will have* $\mathbf{CG}(w_1, w_2) \leq \epsilon$ *for every pair of models* $w_1, w_2$ *on its surface.*

*Proof.* Recall the definition of convexity gap as the maximum value of the barrier height of any segment $\theta_1, \theta_2$ along the interpolation between $w_1$ and $w_2$ from Equation 4:

$$\mathbf{CG}(w_1, w_2) = \sup_{\gamma, \beta} \mathbf{BH}(\gamma w_1 + (1 - \gamma)w_2, \beta w_1 + (1 - \beta)w_2) \qquad \gamma, \beta \in [0, 1] \qquad (5)$$

As we are in an $\epsilon$-basin, the defining inequality from Equation 2 holds $\forall \theta_1, \theta_2 \in \epsilon-$convex basin :

$$\mathcal{L}(\sum_{k=1}^{n} \alpha_k \theta_k) \leq \epsilon + \sum_{k=1}^{n} \alpha_k \mathcal{L}(\theta_k) \qquad (6)$$

Applying this for $n = 2$:

$$\mathcal{L}(\alpha_1 \theta_1 + (1 - \alpha_1)\theta_2) - (\alpha_1 \mathcal{L}(\theta_1) + (1 - \alpha_1)\mathcal{L}(\theta_2)) \leq \epsilon \qquad \forall \theta_1, \theta_2 \in \epsilon\text{-basin} \qquad (7)$$

Hence the supremum of the quantity on LHS is also $\leq \epsilon$. Seeing the definition of $\mathbf{BH}$ from Equation 3, we immediately see that:

$$\mathbf{BH}(\theta_1, \theta_2) \leq \epsilon \qquad \forall \theta_1, \theta_2 \in \epsilon\text{-basin} \qquad (8)$$

As $\mathbf{CG}$ is the supremum of $\mathbf{BH}$ over elements within the $\epsilon-$convex basin only, we have:

$$\mathbf{CG}(w_1, w_2) \leq \epsilon \qquad \forall w_1, w_2 \in \epsilon\text{-basin} \qquad (9)$$

$\square$

## C    HANS PERFORMANCE ON SUBSEQUENCE HEURISTICS

Models that perform poorly on subsequence heuristics tend to fall in the bag-of-words basin. However, the clusters are less pronounced than for either constituent or lexical overlap heuristics (Fig. 9).

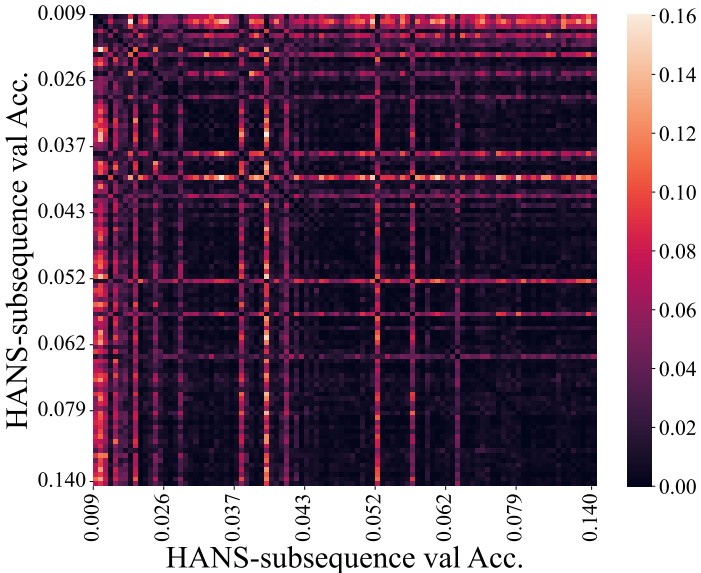

Figure 9: **CG** (over in-domain MNLI validation loss) for MNLI model pairs, sorted by increasing performance on HANS-subsequence.

## D    FLEXIBLE MODE CONNECTIVITY

While our results strongly suggest that linear mode connectivity breaks down between heuristic and generalizing models, they should nonetheless be connected, at least by a nonlinear path of constant training loss (Draxler et al., 2019; Benton et al., 2021). We consider the behavior of HANS generalization along this path, which maintains near-constant ID loss. These paths were easy to identify with Riemannian curves and segmented lines (Garipov et al., 2018), and exhibited poor HANS-LO generalization the further along the path the parameters fall, as shown in Fig. 10.

The results here show an inversion of the pattern exhibited on HANS-LO loss in the linear interpolation case: whereas Fig. 1(a) shows heuristic-to-generalizing connections to exhibit peaks in ID loss but no corresponding spikes in OOD loss, here we find constant-loss paths which exhibit peaks in HANS-LO loss. It appears that gradient descent finds generalizing models in spite of the fact that they exist in a constant-loss multidimensional volume which exhibits a wealth of heuristic models.

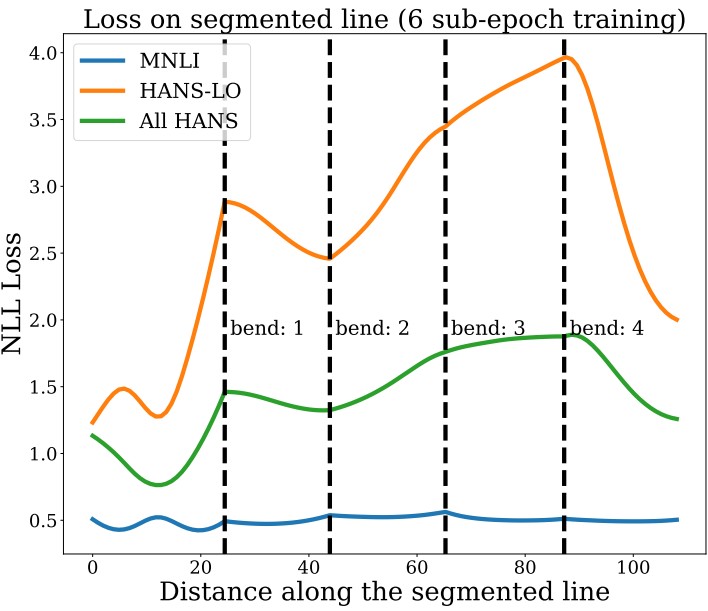

Figure 10: Loss along the segmented curve (after 6 subepochs) between a heuristic and generalizing NLI model. 1 sub-epoch is equivalent to $\sim 120$ training steps with a batch size of $128$, and a learning rate of $8 \times 10^{-5}$.

## E    CORRELATIONS BETWEEN PERFORMANCE ON HANS SUBSETS

As provided in Fig. 11, performance on HANS-LO non-entailing, which reflects an aversion to the lexical overlap heuristic, is positively correlated with ID accuracy. However, in the case of HANS-constituent a model that successfully averts the constituent overlap heuristic is likely to perform poorly ID and on most diagnostic sets. We therefore conjecture that models which avoid constituent overlap heuristics are likely failing to acquire syntax, rather than effectively acquiring better generalization strategies.

Performance on the subsequence heuristic diagnostic set is positively correlated with performance on both LO and constituent heuristic diagnostic sets, as it includes aspects of both other heuristics. For every diagnostic set heuristic, performance on the non-entailing subset (i.e., examples that violate the heuristic) is negatively correlated with performance on the entailing subset (i.e., examples that adhere to the heuristic).

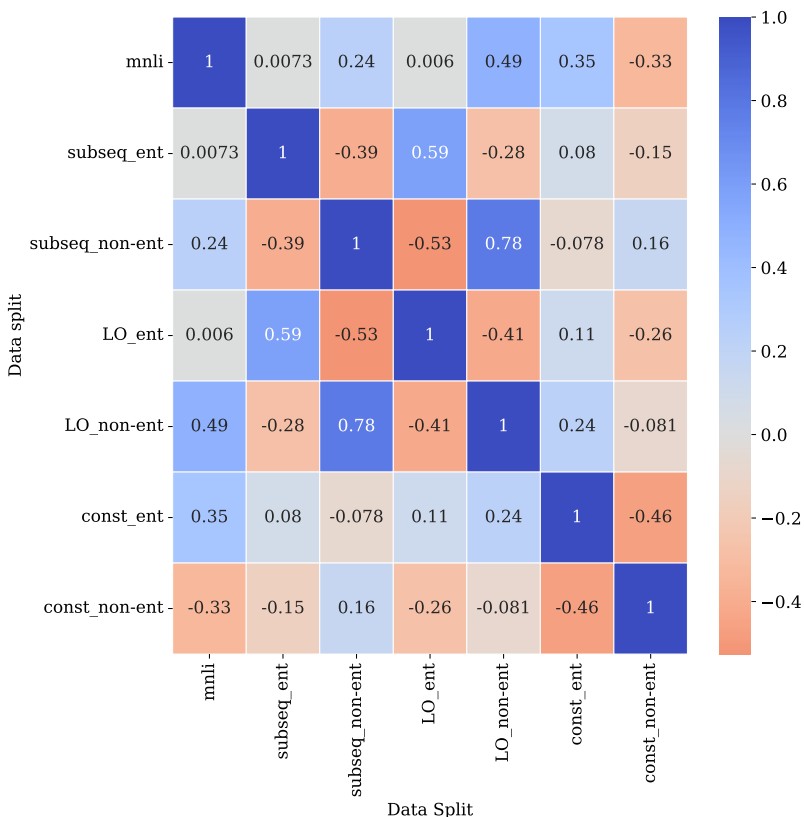

Figure 11: Correlation between accuracy on various ID and OOD diagnostic NLI test sets. MNLI refers to the MNLI validation set. Other sets were taken from HANS validation sets for subsequence, constituent, and lexical overlap heuristics. Both entailing and non-entailing subsets are included.

## F    EXPERIMENTAL DETAILS

### F.1    FINETUNING THE MODELS

The QQP models were trained for 3 epochs, with a learning rate of $2 \times 10^{-5}$, a batch size of 32 samples and a weight decay of $0.01$ from the `bert-base-uncased`[10] pre-trained checkpoint using the google script.[11]  This script uses the AdamW (Loshchilov and Hutter, 2017) optimizer with a warm-up during the first $10\%$ of the training followed by a linear schedule for weight decay. Because these hyperparameters are the recommended defaults for BERT training, they were also used by McCoy et al. (2020) to train the MNLI model set we analyze.

### F.2    INTERPOLATIONS

In order to evaluate the interpolation between a pair of models, we sample a random set of 512 samples from the target dataset and then test each of the interpolated models on this same sample. For an interpolation between two fine-tuned models $\theta_1$ and $\theta_2$, we evaluate the models with interpolation coefficients $\alpha$ (Eqn 1) at intervals of $\frac{1}{10}$ on the line joining $\theta_1$ and $\theta_2$. The resulting plots are shown in Fig. 1(a) and Fig. 1(b). We calculate the convexity gap $\mathbf{CG}(\theta_1, \theta_2)$ is using the loss values of these interpolated models.

---

[10] https://storage.googleapis.com/bert_models/2018_10_18/uncased_L-12_H-768_A-12.zip
[11] https://github.com/google-research/bert

In order to gauge the effect of number of samples, we repeated interpolation experiments on MNLI using only 256 samples (instead of 512). Using a smaller sample reduced the magnitude of correlation coefficient in Fig. 3(a) to $\rho = -0.65$, but the cluster assignments remain the same.

### F.3 MODEL EVALUATIONS

When evaluating the performance of a finetuned model (e.g., to sort the models on the linear connectivity heatmaps, or to select the top and bottom 5 models for Fig. 1(a) and Fig. 1(b)), we use the whole dataset available. These include the entirety of:

- All non-entailing samples of a given heuristic from the `test` split of HANS.
- The `dev_and_test` split of PAWS-QQP.
- The entire QQP and MNLI validation sets.

As Feather-BERTs have been finetuned on MNLI, to evaluate them on HANS tasks, we would need to convert the logits for the three MNLI classes (`entailment`, `neutral`, `contradiction`) to logits for the two HANS classes(`entailing` and `non-entailing`). We convert these labels by taking the logit for `non-entailing` class as the maximum of the logits of the MNLI `contradiction` and `neutral` classes, and taking the logit for `entailing` class of HANS as the logit of `entailing` class of MNLI.

### F.4 COMPUTING SHARPNESS

For calculating $\epsilon$-sharpness of a model, we use the definition of Keskar et al. (2017):

$$\phi_{x,f}(\epsilon, A) := \frac{\max_{y \in \mathcal{C}_\epsilon}(f(x + Ay)) - f(x)}{1 + f(x)} \times 100 \tag{10}$$

where $f$ is the loss function, $x \in \mathbb{R}^n$ are the original parameters of the model, $A \in \mathbb{R}^{n \times p}$ is a matrix that restricts the calculation of $\epsilon-$sharpness to a subspace of the parameter space, and

$$\mathcal{C}_\epsilon = \{z \in \mathbb{R}^p : -\epsilon(|(A^+x)_i| + 1) \leq z_i \leq \epsilon(|(A^+x)_i| + 1) \forall i \in [p]\} \tag{11}$$

Following the defaults in Keskar et al. (2017), we take $A$ as $\mathbb{I}_{n \times n}$. Using 32768 samples from MNLI `validation_matched`, we evaluate the initial loss $f(x)$. We use the same samples to compute the maximum loss $\max_{y \in \mathcal{C}_\epsilon} f(x + Ay)$, using an SGD optimizer with a learning rate of $8 \times 10^{-5}$, a batch size of 32, and accumulating gradient over 4 batches. In order to compute maximum loss, we perform a total of 8192 gradient updates, each followed by clamping of the weights within $\mathcal{C}_\epsilon$. We set $\epsilon$ to $1 \times 10^{-5}$.

#### F.4.1 COMPUTATIONAL RESOURCES

Finetuning 100 QQP models cost around 500 GPU-hours and 48 CoLA models cost around 10 GPU-hours. Each $100 \times 100$ interpolation and evaluation consumed 114 GPU hours; these were performed for both QQP (at three stages during finetuning, for Fig. 7) and MNLI. The $48 \times 48$ interpolation for CoLA cost about 28 GPU-hours. The experiments add up to a total cost of approximately 994 GPU-hours on a mix of NVIDIA RTX8000 and V100 nodes.

## G TRAINING LOSS

Even without any validation or test set to analyze, we can use the training set alone to make predictions about a model's generalization strategies. Fig. 12 shows a strong relation between training loss convexity gaps and performance on HANS-LO. Future work can develop methods for predicting generalization early based entirely on training behavior.

## H IN-DOMAIN GENERALIZATION

It is well known that OOD generalization is strongly correlated with ID performance in general (Miller et al., 2021). Let us therefore consider the skeptical position that basin membership does not

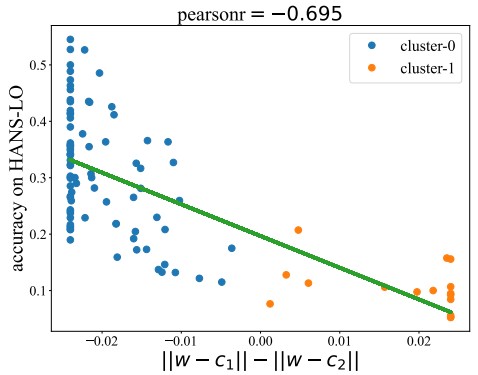

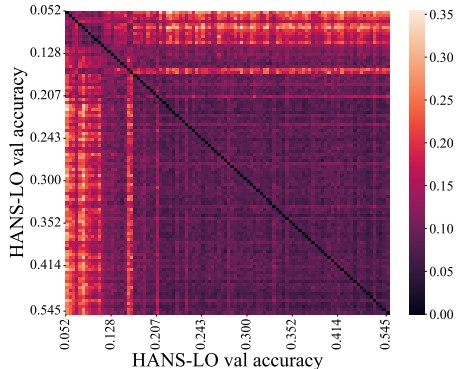

(a) Clusters are based on convexity gaps in the MNLI training loss surface. Best squares fit shown.

(b) Color indicates convexity gap. Models on axes are sorted according to OOD performance.

Figure 12: Relationship between **CG** on the MNLI *training loss surface* and accuracy on the HANS-LO diagnostic set.

| train | test | $f$ | $\mu_{C_1}$ | $\sigma_{C_1}$ | $\mu_{C_2}$ | $\sigma_{C_2}$ | $\frac{\mu_{C_1}-\mu_{C_2}}{\sigma_{C_1}}$ | $\frac{\mu_{C_1}-\max_{w \in C_2} f(w)}{\sigma_{C_1}}$ |
|---|---|---|---|---|---|---|---|---|
| MNLI | MNLI | acc | 0.844 | 0.002 | 0.839 | 0.002 | 2.50 | 1.00 |
| MNLI | HANS-LO | acc | 0.301 | 0.110 | 0.096 | 0.033 | 1.86 | 1.30 |
| QQP | QQP | F1 | 0.879 | 0.001 | 0.872 | 0.002 | 7.00 | 5.00 |
| QQP | PAWS-QQP | F1 | 0.436 | 0.005 | 0.424 | 0.004 | 2.40 | 0.60 |
| CoLA | CoLA ID | MCC | 0.601 | 0.017 | 0.600 | N/A | 0.06 | 0.06 |
| CoLA | CoLA OOD | MCC | 0.537 | 0.015 | 0.576 | N/A | -2.60 | -2.60 |

Table 1: Cluster mean, standard deviation, and distribution overlap for all tasks on both ID and OOD performance under metric $f$. $C_1$ is the larger cluster on each task. On CoLA, $C_2$ contains only a single model.

directly relate to OOD generalization strategies, but instead OOD performance is mediating the effect of general model quality in-domain. While ID (on MNLI) and OOD (on HANS) performance are certainly positively correlated (Appendix E), we find that all 12 of the models in the syntax-unaware cluster fall at least 1.3 standard deviations below the mean of the syntax-aware cluster's performance on HANS-LO (Table 1), and only 1 standard deviation below the mean on MNLI validation. The partitions are also less clear in the ID metrics (Fig. 13) compared to lexical overlap. Therefore, considered as a binary class, basin membership is more predictive of OOD behavior than of ID on NLI. Furthermore, the fact that performance on HANS-constituent is *negatively* correlated with ID performance (Appendix E) makes it clear that the underlying difference between these clusters is syntax-awareness, not overall model quality.

CoLA presents even stronger evidence for interpreting basin membership as related to OOD generalization over ID. The single outlier model falling outside of the main basin is a full 2.6 standard deviations outside the mean on OOD performance, but only 0.6 on ID performance.

However, we should not dismiss the relationship between ID accuracy and cluster membership. Compared to the diagnostic set PAWS, the ID validation set is more predictive of **CG** on QQP (Fig. 14). In both MNLI and QQP, the relationship between ID and centroid distance also seems more linear than OOD. Based on these findings, the human-defined heuristics of the PAWS dataset may not be the correct characterization of the difference in generalization behaviors between different basins. We leave the discovery of better characterizations of difference in strategy between basins to future work.

One caveat is the fact that ID evaluation and ID convexity gap measurements are performed on the same validation set, because the test sets are not public for these datasets. Therefore, ID performance

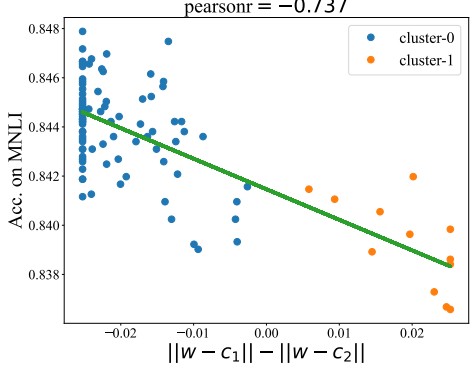

(a) Clusters are based on convexity gaps in the MNLI validation loss surface. Best squares fit shown.

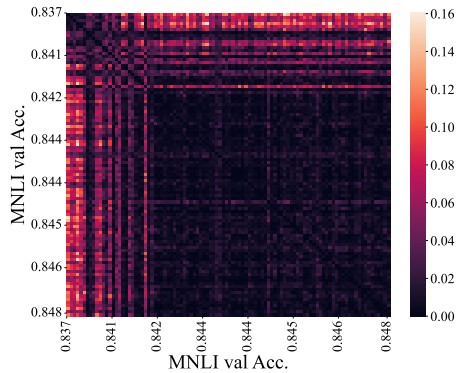

(b) Color indicates convexity gap. Models on axes are sorted according to ID performance.

Figure 13: Relationship between **CG** on the MNLI in-domain validation loss surface and MNLI in-domain validation accuracy.

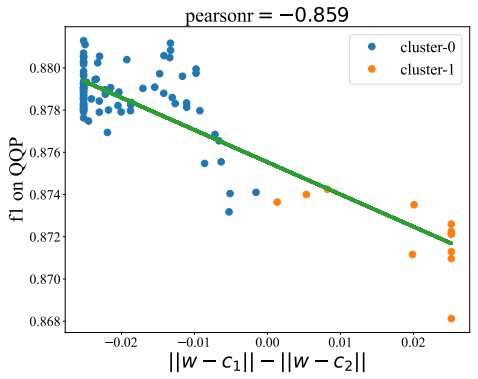

(a) Clusters are based on convexity gaps in the QQP validation loss surface. Best squares fit shown.

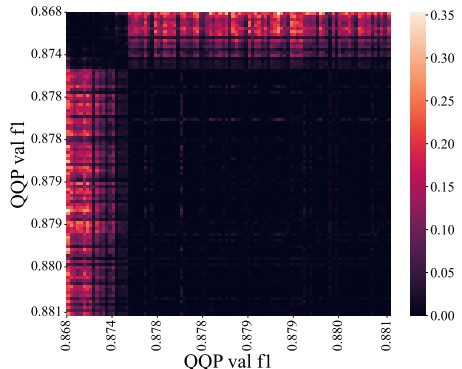

(b) Color indicates convexity gap. Models on axes are sorted according to ID performance.

Figure 14: Relationship between **CG** on the QQP in-domain validation loss surface and QQP in-domain validation accuracy.

is exposed during the clustering process, casting some doubt on alignment between the clusters and ID performance.

## I  ALTERNATIVE BARRIER MEASUREMENTS

In Fig. 15 we show NLI model clustering using the original barrier height metric (Equation 3) from Entezari et al. (2021). We can see that the Pearson's correlation coefficient is $-0.49$, which is far lower in magnitude than correlation in the **CG**-metric space.

We also show the results for another metric, viz. area under the interpolation curve (AUC), in Fig. 16. Although this shows the same Pearson's correlation coefficient, the clusters are much less crisp in the heatmap. In order to compute AUC, we first subtract the lowest point on the curve from all points, and then compute the area under the shifted curve.

Finally, we consider the effect of Euclidean distance (Fig. 18 and 17). The clustering effect is extremely strong and predictive of generalization behavior. It is worth considering the possibility that basins trap the models in a way that forces them to have larger Euclidean distances, but those Euclidean distances are the property that most determines the generalization strategy of the model.

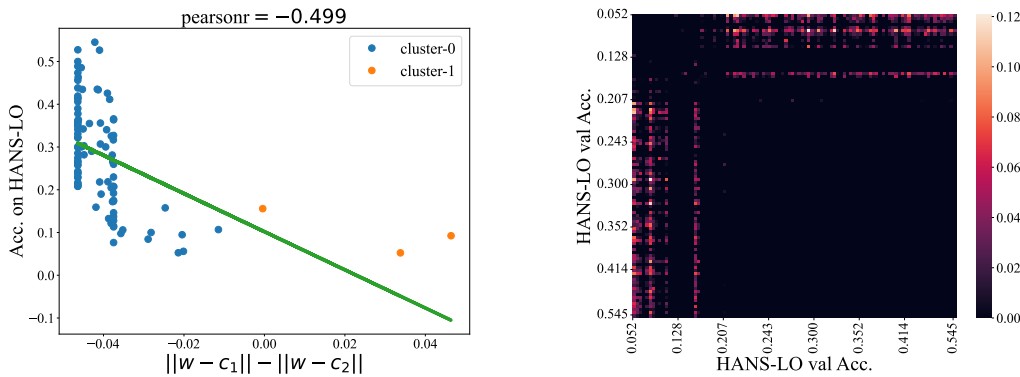

Figure 15: Relationship between **BH** on the MNLI validation loss surface and HANS-LO accuracy.

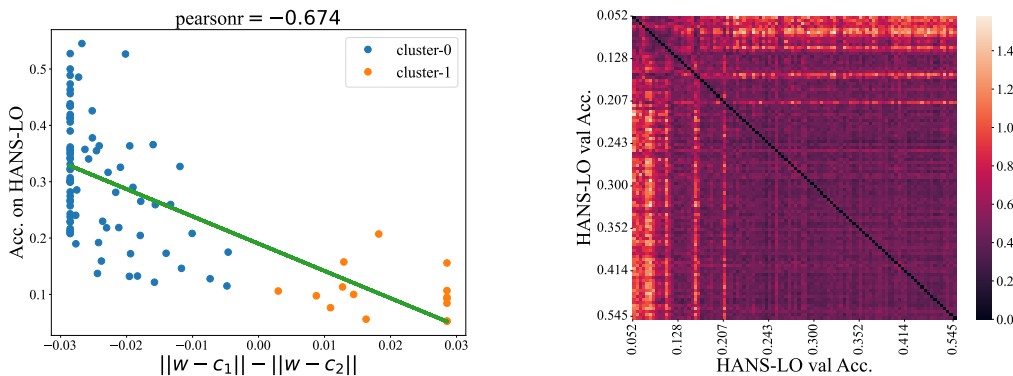

Figure 16: Relationship between AUC on the MNLI validation loss surface and HANS-LO accuracy.

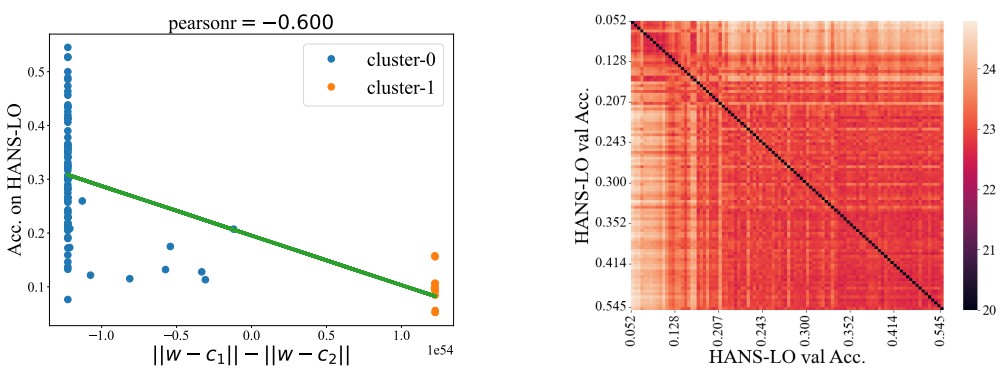

Figure 17: Relationship between Euclidean distance on the MNLI validation loss surface and HANS-LO accuracy.

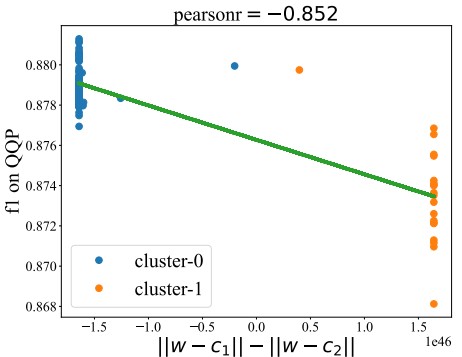 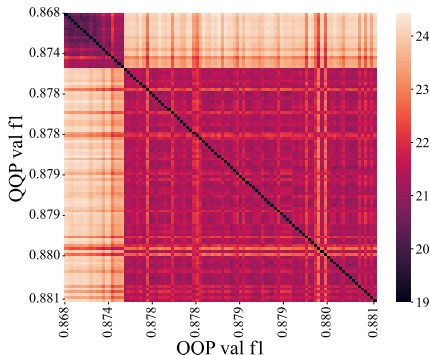

Figure 18: Relationship between Euclidean distance on the QQP validation loss surface and PAWS-QQP accuracy.

## J ROBERTA RESULTS

BERT shows consistent clustering behavior, consistently aligned with generalization performance, across each text classification task measured. RoBERTa (Liu et al., 2019) uses the same architecture but trained longer and with slightly different objective and hyperparameters. Despite this similarity, RoBERTa shows significantly better generalization on HANS-LO (Fig. 19(b)) and does *not* exhibit clear clustering behavior (Fig. 19(a)). These results suggest that the simple basin selection behavior we observe in BERT may indicate that a model can still benefit from longer periods of pretraining.

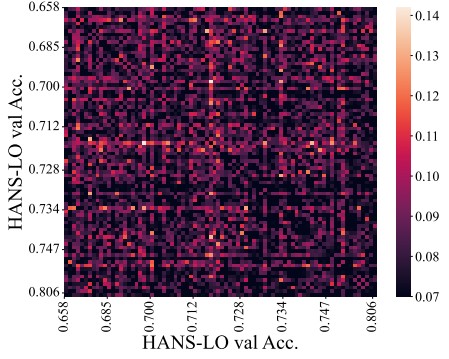 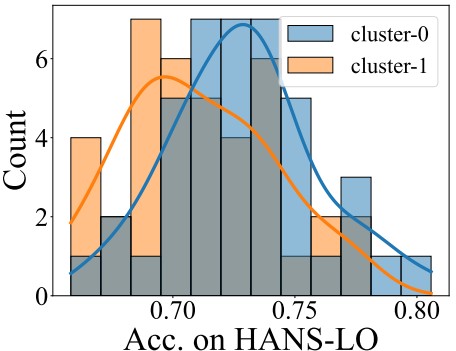

(a) Heatmap of **CG** for finetuned RoBERTa models, sorted by HANS-LO performance. No significant clustering effects are visible, but the scale of barriers is similar to that among the heavily clustered BERT models.

(b) Histogram on HANS-LO performance for finetuned RoBERTa models, with blue and orange indicating the respective models found in each of the two clusters formed by k-means clustering on the **CG** spectral space.

Figure 19: **CG** results on a set of MNLI models finetuned from RoBERTa.

## K RELATION TO FINETUNING DYNAMICS

Kumar et al. (2022) analytically proved that finetuning creates a distorting effect by rescaling existing features rather than learning new features. To remedy this distortion, they proposed that rather than finetuning the entire network at once, we first train the linear classifier head in isolation and then finetune the rest of the network. This procedure is called linear probing then full fine-tuning (LP-FT). We find that LP-FT reduces the range of convexity gaps, and accompanying clustering

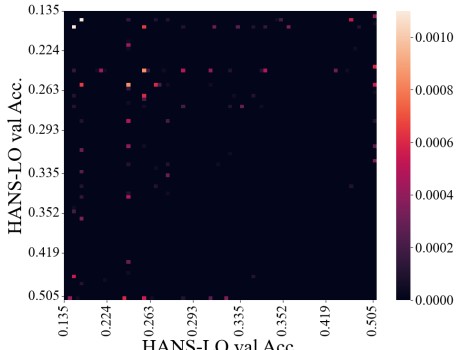
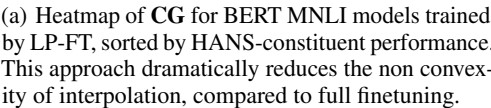
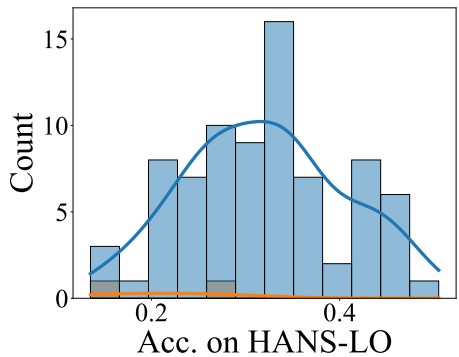

(a) Heatmap of **CG** for BERT MNLI models trained by LP-FT, sorted by HANS-constituent performance. This approach dramatically reduces the non convexity of interpolation, compared to full finetuning.

(b) Histogram on HANS-LO performance for LP-FT BERT models, with blue and orange indicating the respective models found in each of the two clusters formed by k-means clustering on the **CG** spectral space.

Figure 20: **CG** results on a set of MNLI models, initialized from BERT, that have been trained by LP-FT (Kumar et al., 2022).

effects, between models (Fig. 20(a)). This change is accompanied by a slight shift in the distribution of behavior on HANS-LO (Fig. 20(b)).

These findings are to be expected, as training the linear layer with the rest of the model frozen is a convex optimization problem, and therefore each training run should be settling into the same basin to start. These results imply that avoiding certain basins may be part of the advantage of LP-FT over traditional finetuning. Although the resulting models still contain a slight outlier, the strong multi-cluster behavior is no longer evident. It also presents a potential disadvantage of LP-FT for ensembling, however, as the resulting models are more homogenous and therefore ensembling can provide limited benefits.

## L  THE RELATIONSHIP BETWEEN EUCLIDEAN DISTANCE AND CG

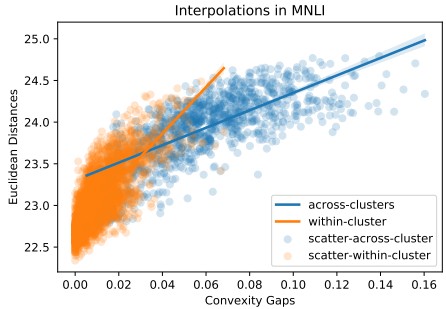
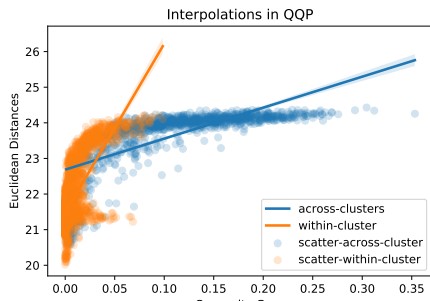

Figure 21: Relationship between Euclidean distance and **CG**. Line of best fit shown separately for connections between and within basin clusters found in the validation loss **CG** distance space. (a) MNLI models (b) QQP models

One extreme possibility is that these are not distinct basins, but instead that every model falls on the perimeter of a single radially symmetric barrier where every cross section is a nearly identical shape. The **CG** would then be determined entirely by Euclidean distance. Although clearly loss surface geometry causes Euclidean distances to emerge rather than vice versa, we also see from Fig. 21 that the relationship between these two metrics is different within vs between basin clusters. Although

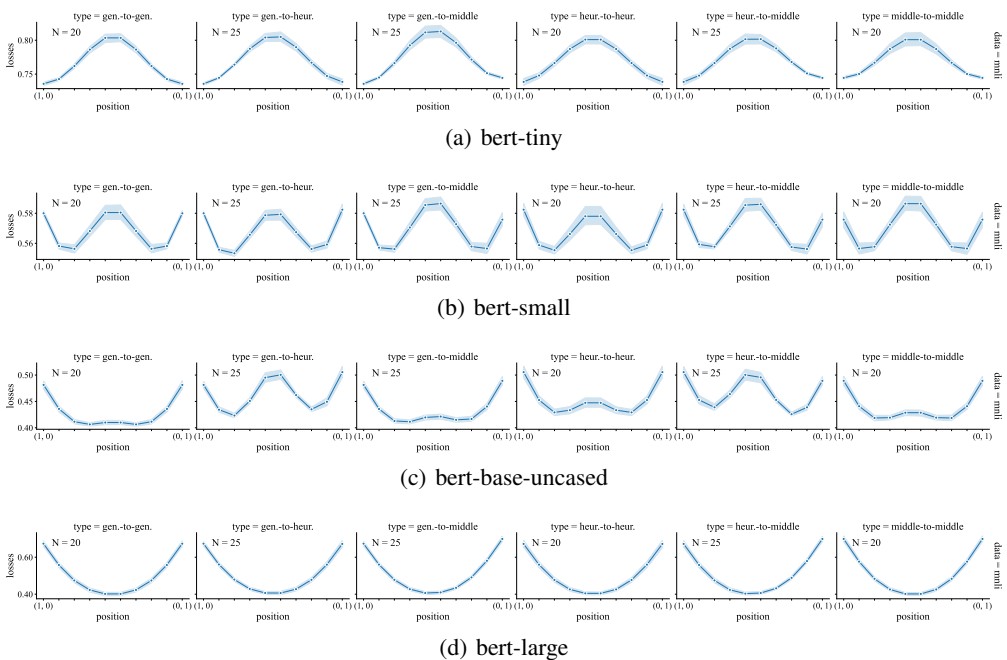

(a) bert-tiny

(b) bert-small

(c) bert-base-uncased

(d) bert-large

Figure 22: **[Loss surface on the line joining pairs of models trained on MNLI.]** Performance during linear interpolation between pairs of models taken from the 5 best (gen.) and 5 worst (heur.) in HANS-LO accuracy and 5 average models (middle). Overall the trend is towards better connectivity as the model size increases.

there is no stark division between connections within and between basins, in general Euclidean distance is more responsive to (i.e., has a greater slope with respect to) the size of the barrier within clusters. Therefore placement of a model within a basin might determine Euclidean distance in a different manner than placement with respect to other basins.

## M  SCALING LAWS OF LINEAR MODE CONNECTIVITY

We explore the effect of model size on linear connectivity of finetuned models in Figure 22. As the scale increases we see the loss landscape changes from one of complete disconnectivity(Figure 22(a)) to one of complete connectivty(Figure 22(d)). This is in line with the results of Ainsworth et al. (2022), which show increasing connectivity for wider ResNet models.

We can also observe that the region closer to the finetuned models becomes low-loss(Figure 22(b)) first, and only as we increase the scale further, does the non-convexity disappears and the region between models become low-loss too. It is as though the basins in which the finetuned models land get progressively closer to each other with increasing model size.

