# OpenReview forum: "Linear Connectivity Reveals Generalization Strategies"
_ICLR.cc/2023/Conference — ICLR 2023 poster_

### Official Review · Reviewer_b2ea · 2022-10-13

**Confidence:** 4
**Correctness:** 3
**Technical Novelty And Significance:** 4
**Empirical Novelty And Significance:** 4
**Recommendation:** 8

**Clarity, Quality, Novelty And Reproducibility:**

The figures and presentation in general are very clear. Code and models are provided. While I did not run the code, the paper overall appears to be very reproducible.

**Strength And Weaknesses:**

Strengths:
- The phenomenon uncovered by this paper is thoroughly investigated and will be of great interest to the community.
- The work is very well contextualized among related investigations.

Weaknesses:
- The experiments are already very thorough, so while it is not necessary I think the paper would be much stronger with the three following experiments.
1. A lot of times in this paper the NLP world is contrasted with the vision world. However, a common technique for fine-tuning in the vision world is to first fine-tune a linear classifier, then unfreeze and fine-tune all params [1]. What happens when you first train a linear probe then use this as the common initial head when fine-tuning pairs - do you still see different generalization strategies? Relatedly, what happens when you start with a T5 or T0 so that you are starting from an already good model, instead of with a new classification head so that performance is initially not good.
2. In vision, LMC also breaks when fine-tuning if you do so with a very high LR. What is the effect of LR in these experiments?
3. Instead of interpolating, what happens when you ensemble the heuristic and generalized models?

[1] https://arxiv.org/abs/2202.10054


**Summary Of The Paper:**

This paper observes barriers in the loss landscape between fine-tune models which have discovered different "generalization strategies" in the context of text classification. In contrast, models with the same generalization strategies have no barrier when interpolating linearly between the two models.

**Summary Of The Review:**

The paper investigates an interesting phenomenon and does so in a clear and reproducible manner. While additional experiments would strengthen the investigations, I believe the paper will be of interest to the community and should be accepted.

---

> ### Author Response · Authors · 2022-11-15
> **Response**
>
> Thank you for your kind words and your suggestions!
>
> **Comment:** A lot of times in this paper the NLP world is contrasted with the vision world. However, a common technique for fine-tuning in the vision world is to first fine-tune a linear classifier, then unfreeze and fine-tune all params [1]. What happens when you first train a linear probe then use this as the common initial head when fine-tuning pairs - do you still see different generalization strategies? Relatedly, what happens when you start with a T5 or T0 so that you are starting from an already good model, instead of with a new classification head so that performance is initially not good.
>
> **Response:** This is actually an experiment that we conduct in Appendix K! We do in fact find that the models show less clustering behavior, although the impact on the distribution of generalization strategies is less straightforward. This makes sense, as the initial linear probing step is now convex, so it is likely that they are able to find the same basin.
>
> We have looked a little at T5, but with randomly initialized heads. On the one task we checked (COGS), we found some mild outlier models
> but no clear clustering behavior (ie, a few models had barriers with several other models, but most models had low CG for every interpolation). It's possible that this kind of outlier behavior can be caused by insufficient data.
>
> **Comment:** In vision, LMC also breaks when fine-tuning if you do so with a very high LR. What is the effect of LR in these experiments?
>
> **Response:** Thank you for making this point about the learning rate. We will try to run these experiments for a final version of the paper, but it’s also something to consider as we do follow up work, because there are many variables that might determine the preference towards particular basins.
>
> **Comment:** Instead of interpolating, what happens when you ensemble the heuristic and generalized models?
>
> **Response:** Ensembling is something that we are really interested in for follow up work! Obviously, intuitively it seems that models with very different generalization strategies should be better to ensemble; we think this direction is likely to become much larger than an appendix experiment, though, as there is a deeper question of when to ensemble by interpolation and when by voting.

---

> > ### Comment · Reviewer_b2ea · 2022-11-15
> > **Thanks**
> >
> > Thank you for the response.

---

### Official Review · Reviewer_XZas · 2022-10-28

**Confidence:** 4
**Correctness:** 3
**Technical Novelty And Significance:** 2
**Empirical Novelty And Significance:** 2
**Recommendation:** 5

**Clarity, Quality, Novelty And Reproducibility:**

#### **Comments on writing**

1. The writing is a bit colloquial in certain places, making it hard for a general audience to read. For instance, it is unclear what the following sentence in the introduction means: "Early LMC may thus predict final heuristics." There is nothing like early LMC; LMC makes sense only when the models have converged. Perhaps the authors should rephrase it to: "Early performance of models on a diagnostic task can reveal information about the model's final behaviors, provided we observe LMC for the particular task." Such writing appears in several places in the paper, and someone unfamiliar with LMC might be unable to make sense of these phrases.

2. I also feel too many abbreviations in the paper make the reading non-smooth. I suggest referring to the diagnostic tasks as "task  1, 2, 3,"  etc. The exact nature of these tasks is relevant only when discussing the linguistic connection and saying something specific about the generalization behavior. For instance, sentences like "HANS performance during interpolation: It is clear from Fig. 1(a) that, when interpolating between LO-heuristic models, HANS-LO loss" can be made much more straightforward. Section 2, in particular, can be made more readable for the general audience, who understand LMC, but might not be too familiar with the NLP tasks.

3. What does this mean: "directly pessimizing on the test set."?












**Strength And Weaknesses:**

The paper asks an interesting question: how does the functional behavior differ between different modes split from a pre-trained model? It discusses several valid and interpretable diagnostic sets for doing this. If the experiments in the paper were performed correctly (see my apprehensions later), it would offer novel insight about LMC.


 #### **Missing important experimental details**
My main issue with the paper is that it is not very precise with the different experimental details. In some cases, it just leads to a non-smooth reading of the paper, but in other cases, it leads to significant confusion about what is being done in the experiments.

1. How were the plots in figure 2 obtained? Are we looking at two random directions, two principal component directions, or something else altogether? One could miss high-dimensional barriers by looking at random directions; non-convex functions can look convex, etc.

2. Since several data sets/tasks are involved, it is crucial to be specific about the loss landscape on which LMC is observed or not observed. I could infer which landscape is being plotted and which models are connected on which task, but only after carefully checking the paper and the appendix. This should be more clearly specified in the figure captions and should not be a matter of confusion. For reference, see section 2 in [this paper](https://arxiv.org/pdf/2008.11687.pdf), and note how non-colloquial markers are used throughout the paper, to be specific.

3. Regarding footnote 3, were different endpoints obtained using different initializations for the linear layer? This seems to be the case since the same seed is used for initialization and data permutation. If so, this is different from the usual LMC setup, where the branches are split from the same training trunk after several training steps from the same initialization. This is a crucial point, as models training from different initializations don't usually show LMC. Most importantly, **this could be why different fine-tuning runs converge to different basins**.


#### **Convexity Gap**

I am less convinced about the convexity gap metric. It is easy to look at figures and enforce one's interpretations. Based on the discussion in the paper, it is unclear if CG dominates other strategies. The clustering in Figure 18 looks comparable to CG. The authors should discuss this in the main paper. In light of this, it is clear that there are clusters as they are visible under all metrics, but it is not clear if CG is the best way to obtain clusters.  Similarly, the discussion in appendix H is critical and can't be pushed to the appendix. It is not clear to me, based on the appendix, that the authors successfully refute the skepticism that clustering is indicative of target domain performance, and the differences can be heightened by looking at diagnostic tasks. Overall, this discussion is not mature yet and requires further investigation.




**Summary Of The Paper:**

The paper studies linear mode connectivity (LMC) for language tasks. Usually, LMC is studied on the training and test loss landscape, both of which come from the same distribution. [Neyshabur et al.](https://arxiv.org/pdf/2008.11687.pdf) study LMC for several fine-tuning tasks, and show that the LMC can be observed on the fine-tuning landscapes when initializing from the same pre-trained model. However, LMC isn't observed when initializing from random initialization. [Neyshabur et al.](https://arxiv.org/pdf/2008.11687.pdf) developed this observation to understand "what is being transferred in transfer learning." They make the case that pretraining helps by allowing LMC on the target landscape and biasing the optimizer trajectories to converge to a shared basin.

In this paper, the authors take this further, and instead of just studying LMC on the target task's landscape, they also consider an out-of-domain diagnostic task. The authors show that while fine-tuned models behave well on the target task, only some of them behave well on the diagnostic task. The models which do well (generalizing models) and do not do well (heuristic models) on the diagnostic tasks form separate clusters on the target landscape. Each of these clusters is mode connected, but they are not connected to each other.  The paper then suggests a notion of "distance" in the model space called "convexity-gap," which leads to meaningful clustering of the models in terms of their functional behavior. Finally, the paper shows that as training proceeds, different models get closer to their clusters, suggesting that early diagnostic tests of models can predict their final generalization performance.

**Summary Of The Review:**

I think the results and discussion in section three are interesting, but I have some concerns about the experimental procedure. I also don't like the writing of the paper, as the authors are not the most explicit about some important details. But most importantly, I believe the metric proposed in section four needs to be investigated further. The experiments don't depict any clear advantage of the metric over simpler measures. Overall, this reduces the novelty of the work. I encourage the authors to make the writing changes, develop more on their metric, add some extensions mentioned in section six, and then resolve future directions to improve the paper.

On an orthogonal note, I am also curious if the authors have explored the connections to neural mode collapse?

---

> ### Author Response · Authors · 2022-11-15
> **Missing details**
>
> Thank you for your suggestions to improve our writing!
>
> **Comment:** How were the plots in figure 2 obtained? Are we looking at two random directions, two principal component directions, or something else altogether? One could miss high-dimensional barriers by looking at random directions; non-convex functions can look convex, etc.
>
> **Response:** The directions in figure 2 are defined strictly by the model weights as follows: Through three points in $\mathbb{R}^n$, a unique plane passes, no matter how big $n$ is. We observe the loss surface on this plane. Now this plane can be plotted in many ways, each of which would be a rotated version of any other. The X direction is just G1-G0 (no projections) and the Y direction is the vector perpendicular to G1-G0 in the plane containing G0, G1, G2. Dimensionality reduction by selecting principal components is only necessary when more than 3 models are at play.
>
> **Comment:** Since several data sets/tasks are involved, it is crucial to be specific about the loss landscape on which LMC is observed or not observed. I could infer which landscape is being plotted and which models are connected on which task, but only after carefully checking the paper and the appendix. This should be more clearly specified in the figure captions and should not be a matter of confusion. For reference, see section 2 in this paper, and note how non-colloquial markers are used throughout the paper, to be specific.
>
> **Response:** Thank you for this suggestion; We have added details to the captions to help with clarity.
>
> **Comment:** Regarding footnote 3, were different endpoints obtained using different initializations for the linear layer? This seems to be the case since the same seed is used for initialization and data permutation. If so, this is different from the usual LMC setup, where the branches are split from the same training trunk after several training steps from the same initialization. This is a crucial point, as models training from different initializations don't usually show LMC. Most importantly, this could be why different fine-tuning runs converge to different basins.
>
> **Response:** It is true that much of the work on LMC of pretrained models doesn’t use a randomized linear classification layer. However, this is the standard set up for finetuning in practice, and work on weight averaging often uses these more realistic finetuning settings (such as [Choshen et al.](https://arxiv.org/abs/2211.00107)). Because assumptions of linear connectivity and methods based on weight interpolation are inextricably linked, our work sheds light on why such methods often fail in NLP settings.
>
> Furthermore, not all LMC work relies on neuron alignment through synchronized early training or pretraining; more recent work on controlling for permutation (Ainsworth et al.; Entezari et al.) instead assumes completely diverging trajectories from the beginning of finetuning. Our own setting, which is a conventional finetuning setup, falls in between finetuning without new random weights (Neyshabur et al.) and fully random setups (Ainsworth et al.; Entezari et al.), both of which are settings that have shown LMC in image classification tasks. Therefore, it may be surprising that a setting with far less initialization variance than these works fails to show linear mode connectivity.

---

> > ### Author Response · Authors · 2022-11-15
> > **Convexity gap**
> >
> > **Comment:** I am less convinced about the convexity gap metric. It is easy to look at figures and enforce one's interpretations. Based on the discussion in the paper, it is unclear if CG dominates other strategies. The clustering in Figure 18 looks comparable to CG. The authors should discuss this in the main paper. In light of this, it is clear that there are clusters as they are visible under all metrics, but it is not clear if CG is the best way to obtain clusters. Similarly, the discussion in appendix H is critical and can't be pushed to the appendix. It is not clear to me, based on the appendix, that the authors successfully refute the skepticism that clustering is indicative of target domain performance, and the differences can be heightened by looking at diagnostic tasks. Overall, this discussion is not mature yet and requires further investigation.
> >
> >
> > **Response:** Thank you for highlighting Figure 18. We need to clarify that the value of CG is not only in identifying the strategy of a model, but in providing an insight into how the loss surface is structured. Our results stand in contrast to existing work on LMC in that we identify multiple basins and point to semantically meaningful differences between models in these basins. Therefore, even if a different metric is equally predictive of generalization behavior, it does not invalidate the usefulness of CG for analyzing loss surfaces.
> >
> > In the case of Euclidean distance, we actually would predict that Euclidean distance would be highly predictive of mechanistic generalization behavior, given our understanding of basins. This is because the geometry of the loss surface impacts the distance between the models’ final positions. Regardless of which metric is best for clustering behavior, convexity gap is a better explanatory variable for clustering because the initial random weights are not clustered in the euclidean space. Different models drawn to the same attractor basin are, however, likely to be closer together in euclidean space than models whose trajectory is determined by a different attractor. CG gives clear insights into the discrete structure of the loss surface which explains the similar clustering effects in Euclidean space.
> >
> > To completely address this note: One extreme possibility is that these are not distinct basins, but instead that every model falls on the perimeter of a single radially symmetric barrier where every cross section is a nearly identical shape. In this case, the CG would be linearly determined by Euclidean distance. We have some results indicating that the relationship between CG and Euclidean distance is somewhat different within vs between different basins, so this cannot be this case, and have added an appendix (L) to this effect.

---

> > > ### Author Response · Authors · 2022-11-15
> > > **clarity**
> > >
> > > **Comment:** The writing is a bit colloquial in certain places, making it hard for a general audience to read. For instance, it is unclear what the following sentence in the introduction means: "Early LMC may thus predict final heuristics." There is nothing like early LMC; LMC makes sense only when the models have converged. Perhaps the authors should rephrase it to: "Early performance of models on a diagnostic task can reveal information about the model's final behaviors, provided we observe LMC for the particular task." Such writing appears in several places in the paper, and someone unfamiliar with LMC might be unable to make sense of these phrases.
> > >
> > > **Response:** When we refer to “early LMC”, we simply mean the convexity of interpolations between models at earlier checkpoints in fine tuning. Although connectivity is not something that is guaranteed early in training, it still can be measured early in training, and appears to be predictive of the final basin a model ends up in. We have rephrased to clarify.
> > >
> > > **Comment:** I also feel too many abbreviations in the paper make the reading non-smooth. I suggest referring to the diagnostic tasks as "task 1, 2, 3," etc. The exact nature of these tasks is relevant only when discussing the linguistic connection and saying something specific about the generalization behavior. For instance, sentences like "HANS performance during interpolation: It is clear from Fig. 1(a) that, when interpolating between LO-heuristic models, HANS-LO loss" can be made much more straightforward. Section 2, in particular, can be made more readable for the general audience, who understand LMC, but might not be too familiar with the NLP tasks.
> > >
> > > **Response:** Thank you for this suggestion. We prefer to use more memorable labels rather than numerals for the dataset names (as all of the datasets are fairly common, so many readers will recognize the acronyms more easily than a numerical identifier), but have bolded their first occurrence to make it easier to find their definitions. One possible way to help readers who are unfamiliar with the text classification datasets is to add “OOD” to every mention of one of the OOD datasets and “ID” to every mention of the original in-domain datasets. Do you think that this approach might resolve some of the difficulty you experienced when reading?
> > >
> > > **Comment:** What does this mean: "directly pessimizing on the test set."?
> > >
> > > **Response:** “Pessimize” is the opposite of “optimize”, i.e., maximizing the loss. We have added clarifying context (by contrasting with "optimize"), as it is vocabulary that only occasionally appears in the optimization literature.

---

> > > > ### Comment · Reviewer_XZas · 2022-12-05
> > > > **Response**
> > > >
> > > > Apologies for the late response.
> > > >
> > > > > One possible way to help readers who are unfamiliar with the text classification datasets is to add “OOD” to every mention of one of the OOD datasets and “ID” to every mention of the original in-domain datasets. Do you think that this approach might resolve some of the difficulty you experienced when reading?
> > > >
> > > > Yes, the issue is people not too familiar with NLP datasets might also read the paper. This will make it easier for them to understand what is happening in the abstract.
> > > >
> > > > >  “Pessimize” is the opposite of “optimize”, i.e., maximizing the loss. We have added clarifying context (by contrasting with "optimize"), as it is vocabulary that only occasionally appears in the optimization literature.
> > > >
> > > > Not sure it makes any sense; optimization doesn't mean minimization. I would just avoid the term altogether.
> > > >
> > > > > When we refer to “early LMC”, we simply mean the convexity of interpolations between models at earlier checkpoints in fine tuning.
> > > >
> > > > This is indeed confusing, as I thought of something else altogether. Using mode to refer to a checkpoint early in training is unnecessary ambiguity. Perhaps consider saying something like, "the trajectories interpolate."
> > > >
> > > > > We need to clarify that the value of CG is not only in identifying the strategy of a model, but in providing an insight into how the loss surface is structured.
> > > >
> > > > This is unclear to me; if a new metric provides a little more intuition over the existing definition and doesn't have demonstrable predictable power over simple baselines, why introduce it?
> > > >
> > > > > This is because the geometry of the loss surface impacts the distance between the models’ final positions. Regardless of which metric is best for clustering behavior, convexity gap is a better explanatory variable for clustering because the initial random weights are not clustered in the euclidean space.
> > > >
> > > > Yes, but the experiments don't show it is a better metric, even if one would "hope" it is. That is my point.
> > > >
> > > > > We have some results indicating that the relationship between CG and Euclidean distance is somewhat different within vs between different basins, so this cannot be this case, and have added an appendix (L) to this effect.
> > > >
> > > > Thanks, this should be helpful.
> > > >
> > > > > The directions in figure 2 are defined strictly by the model weights as follows...
> > > >
> > > > It would be good to clarify this further in the figure caption and/or text.
> > > >
> > > > > Thank you for this suggestion; We have added details to the captions to help with clarity.
> > > >
> > > > Thanks!
> > > >
> > > > >  It is true that much of the work on LMC of pretrained models doesn’t use a randomized linear classification layer. However, this is the standard set up for finetuning in practice, and work on weight averaging often uses these more realistic finetuning settings (such as Choshen et al.).
> > > >
> > > > It will be good to clarify this subtle point explicitly and mention this work.
> > > >
> > > > > Because assumptions of linear connectivity and methods based on weight interpolation are inextricably linked, our work sheds light on why such methods often fail in NLP settings.
> > > >
> > > > Not sure what the point is here.
> > > >
> > > > > Furthermore, not all LMC work relies on neuron alignment through synchronized early training or pretraining; more recent work on controlling for permutation (Ainsworth et al.; Entezari et al.) instead assumes completely diverging trajectories from the beginning of fine-tuning.
> > > >
> > > > The point of those papers is to show how those diverging strategies can be made to interpolate through permutations, etc. So of course, they don't consider settings where LMC already happens. But the point of this paper is to understand what happens semantically when LMC happens. So it is important to stick to simple setups used in practice. My entire point was to use the same linear layer initialization to not conflate the several aspects of optimization here with LMC.
> > > >
> > > > > Therefore, it may be surprising that a setting with far less initialization variance than these works fails to show linear mode connectivity.
> > > >
> > > > Again I am unsure about the purpose of the paper. If the authors think this is the paper's contribution, it certainly isn't discussed. Based on the current writing, the purpose is to understand semantically what makes different modes different and how that difference evolves. I fail to understand why the authors are defending an orthogonal experimental detail to this message. Just make the two training phases as comparable as possible; that resolves everything.
> > > >
> > > >
> > > > I am inclined to keep my score, as I am not convinced of the paper's main contribution. I will wait for the author's response.

---

> > > > > ### Author Response · Authors · 2022-12-07
> > > > > **response**
> > > > >
> > > > > > Yes, but the experiments don't show it is a better metric, even if one would "hope" it is. That is my point.
> > > > >
> > > > > We are specifically engaging with the literature on mode connectivity in our work. While our findings are useful for work on clustering models, our primary objective is to test whether particular claims about LMC are true in more realistic settings outside of image classification. The goal is not to find a better metric, but to test LMC claims, which appear not to hold and have a clear connection to mechanistic generalization behavior.
> > > > >
> > > > > We also do find in the appendix that there is a difference in the relationship between CG and Euclidean distance, depending on whether you are looking within or between clusters (or to put it another way: looking at low CG or high CG).
> > > > >
> > > > > > But the point of this paper is to understand what happens semantically when LMC happens. So it is important to stick to simple setups used in practice.
> > > > >
> > > > > We are in part addressing the literature that claims LMC even under different weight initializations, and we find that in realistic NLP settings, even permutation doesn't change the fact that there are multiple basins which exhibit different behaviors.
> > > > >
> > > > > However, we understand that you are sticking with your score.

---

### Official Review · Reviewer_rtsz · 2022-10-29

**Confidence:** 3
**Correctness:** 3
**Technical Novelty And Significance:** 3
**Empirical Novelty And Significance:** 3
**Recommendation:** 8

**Clarity, Quality, Novelty And Reproducibility:**

- Parts of the work were difficult to parse because of the number of experiments run on different datasets, some of which were used for training, indicating an in-distribution loss landscape, and some were out-of-distribution. The clarity should be improved by clearly indicating which datasets were used for fine-tuning (MNLI and QQP) for each figure. This gives the reader a better idea of what they should expect to see and what occurs.

- How were the heuristic and generalizing models created? It's unclear whether these models are only different based on random initialization seed and some fine-tuned models generalized while others did not. If this is the case, can the authors comment on the conclusion in the referenced paper they mention (Zhou et al. 2020), which implies that this is more of an issue with the diagnostic dataset rather than the model?

**Strength And Weaknesses:**

Strengths:

- Demonstrates that the LMC phenomenon is not universal across the natural language modality.
- Useful observations in the association of LMC with generalization
- Clustering analysis using the CG metric appears to be novel and an alternative to sharpness.

Weaknesses:

- Not much investigation of the architecture dependence. For instance, Ainsworth 2022 showed that the LMC phenomenon depended on the neural network's architecture and width.
- Unclear how pre-training initialized LMC and LMC with random initialization modulo permutations are the same minima.
- Since the authors demonstrate a lack of connectivity on out-of-distribution data, other hyperparameters may impact results. Ainsworth et al. 2022 noted that the LMC phenomenon does not occur until training has converged to some degree. (See Figure 3 at https://arxiv.org/pdf/2209.04836.pdf). Convergence on in-distribution may not indicate convergence on out-of-distribution. How do the authors define the training criterion to ensure the out-of-distribution has sufficiently "converged"?
- Doesn't the clustering analysis of the CG metric require the diagnostic dataset, whereas sharpness does not? If this is the case, the claim about sharpness being inferior seems overstated. While the overall results of the paper would indicate otherwise, has the clustering of the CG approach been tried on MNLI and QQP?

Comments:

- Absence of evidence is not evidence of absence. The LMC hypothesis is not easily refuted empirically because a permutation search procedure's inability to find a viable permutation for LMC does not mean a permutation does not exist. This is why I do not consider it a weakness of this work, but it would be good for the authors to comment on this issue in the manuscript.

**Summary Of The Paper:**

This work explores the linear mode connectivity (LMC) feature of neural networks in the context of language models. The authors observe a contrast with the literature, which claims that (LMC) is nearly ubiquitous in neural networks. The authors note that the empirical findings based on these claims are biased toward the image domain and demonstrate this is not necessarily the case in natural language settings. Furthermore, the authors also find that out-of-distribution generalization is not associated with the sharpness of the minima but the density of models in a particular linearly connected basin.

**Summary Of The Review:**

Overall, this work appears to analyze LMC from the perspective of generalization on OOD data and find previously unknown connectivity patterns and energy barriers. While there are more questions created in the analysis than answers (see above), this work is a good contribution to the LMC story and why the phenomenon might not be as straightforward as previously thought.

---

> ### Author Response · Authors · 2022-11-15
> **Weaknesses**
>
> Thank you for your kind words about our results and for year constructive criticism.
>
> **Comment:** “Not much investigation of the architecture dependence. For instance, Ainsworth 2022 showed that the LMC phenomenon depended on the neural network's architecture and width.”
>
> **Response:** We agree that analyzing the effect of architecture on basin selection behavior would be a good avenue for future research using our methods. We plan to continue investigating the many factors determining priors over basins, but the compute required to train 50-100 models from, for example, BERT-large makes these experiments difficult to complete in only a week. Do you consider this a necessary question to fit into the scope of the current paper? If so, we can run those experiments probably before the end of the discussion period and certainly before the camera ready deadline.
>
> **Comment:** “Unclear how pre-training initialized LMC and LMC with random initialization modulo permutations are the same minima.”
>
> **Response:** If we understand the question, you’re asking why we find the same minima regardless of whether we permute? It seems, from Neyshabur et al., that pretraining results in initializations where the neurons are already pre-aligned, removing the need for permutation.
>
> **Comment:** “How do the authors define the training criterion to ensure the out-of-distribution has sufficiently "converged"?”
>
> **Response:** This is an interesting question. Grokking phenomena indicate that generalization can occur well after convergence; it is possible that even the most typical heuristic models eventually would transition basins, and such transitions might be an explanation for grokking. However, as we point out below, this disconnectivity between basins occurs even on the in-domain loss; we only use OOD data to study the difference between the mechanistic behavior of the models in each basin, so we are studying connectivity on the same sort of in-domain surface that Ainsworth et al. focus on, only on NLP tasks.
>
> **Comment:** “Doesn't the clustering analysis of the CG metric require the diagnostic dataset, whereas sharpness does not?”
>
> **Response:** It seems we need to clarify the data on which we compute CG. The figures in the main body all show CG only on the in domain validation set loss surface; appendix G shows that we can even derive these clusters from CG on the training set itself (though not as sharply). We use OOD loss to study the differences in mechanistic behavior of the models, not to study the loss surface. (Except for the bottom of Figure 1, which illustrates interpolation on the OOD loss surface, while the top of Fig 1 illustrates interpolation on the in domain validation set.) We have added this detail to the captions of figures, as multiple reviewers lost track of what datasets we were interpolating over.

---

> > ### Author Response · Authors · 2022-11-15
> > **Comments**
> >
> > **Comment:** “Absence of evidence is not evidence of absence. The LMC hypothesis is not easily refuted empirically because a permutation search procedure's inability to find a viable permutation for LMC does not mean a permutation does not exist. This is why I do not consider it a weakness of this work, but it would be good for the authors to comment on this issue in the manuscript.”
> >
> > **Response:** While it may not be possible to empirically refute the LMC, we believe that the association between generalization behavior and the apparently different in-domain basins provides a strong validation of the idea that these basins are meaningfully distinct, and not just permutations of the same basin. However, we have added and aside to the effect to thad better permutations might still exist.

---

> > > ### Author Response · Authors · 2022-11-15
> > > **Clarity, Quality, Novelty And Reproducibility:**
> > >
> > > **Comment:** Parts of the work were difficult to parse because of the number of experiments run on different datasets, some of which were used for training, indicating an in-distribution loss landscape, and some were out-of-distribution. The clarity should be improved by clearly indicating which datasets were used for fine-tuning (MNLI and QQP) for each figure. This gives the reader a better idea of what they should expect to see and what occurs.
> > >
> > > **Response:** Thank you for this suggestion. We have revised the captions to specify the models and data that the loss surface describes.
> > >
> > > **Comment:** How were the heuristic and generalizing models created? It's unclear whether these models are only different based on random initialization seed and some fine-tuned models generalized while others did not. If this is the case, can the authors comment on the conclusion in the referenced paper they mention (Zhou et al. 2020), which implies that this is more of an issue with the diagnostic dataset rather than the model?
> > >
> > > **Response:** We have addressed this in our top level comment now, and emphasize it earlier in the paper. The only difference between these models is the random initialization seed for the linear classifier head and data order.

---

### Official Review · Reviewer_bPGJ · 2022-11-01

**Confidence:** 3
**Correctness:** 3
**Technical Novelty And Significance:** 3
**Empirical Novelty And Significance:** 3
**Recommendation:** 6

**Clarity, Quality, Novelty And Reproducibility:**

The paper is well-written however there are unclear points regarding the experimental setup (see above).

**Strength And Weaknesses:**

The empirical observations are interesting and show novel connections between a model's behavior (in terms of its out-of-domain generalization and preferences).

The work proposes new metrics that seem intuitive and are suitable for what they're designed to capture, like the convexity gap, epsilon-basin.

The main weakness, in my opinion, is that there seems to be lacking information regarding the experiments. Most, if not all, experiments rely on having 'heuristic' and 'generalizing' models, but I couldn't find details on how these models are obtained. Are generalizing and heuristic models trained the exact same way and only differ in the random seed? If different training settings were used, then a detailed discussion on it would be very valuable.

Although the paper shows interesting observations, it's unclear whether these have meaningful implications and/or applicability, and are somewhat restricted to the NLP tasks explored.

**Summary Of The Paper:**

The paper studies linear connectivity from a perspective of out-of-domain generalization for NLP tasks. Experiments show that 'generalizing' and 'heuristic' models (which are defined based on their out-of-domain performance) are separated in terms of what 'basins' they belong to. Further analysis along with clustering techniques show a connection between models' behaviors and their position in the loss surface.

**Summary Of The Review:**

To reiterate from 'strengths and weaknesses':

- Novel and interesting observations.
- Well-defined metrics.
- Lacking some information on experimental setup.
- Limited implications.

---

> ### Author Response · Authors · 2022-11-15
> **Response**
>
> Thank you for your interest in our empirical observations.
>
> **Comment:** The main weakness, in my opinion, is that there seems to be lacking information regarding the experiments. Most, if not all, experiments rely on having 'heuristic' and 'generalizing' models, but I couldn't find details on how these models are obtained. Are generalizing and heuristic models trained the exact same way and only differ in the random seed? If different training settings were used, then a detailed discussion on it would be very valuable.
>
> **Response:** All models are indeed trained identically except for using different random seeds for the linear classifier head initialization and the data shuffle. We have added a more emphatic explanation of the source of variance, as it was missed by multiple reviewers. The generalizing / heuristic breakdown emerges naturally from where the models fall on the loss surface and their corresponding generalization behaviors, not from any deliberate introduction of heuristics.
>
> **Comment:** Although the paper shows interesting observations, it's unclear whether these have meaningful implications and/or applicability, and are somewhat restricted to the NLP tasks explored.
>
> **Response:** We recognize that our results are only presented for a small number of text classification tasks, which were specifically selected because they each had a common OOD challenge set on which to test generalization behavior. However, existing results on the topic of LMC are confined almost entirely to image classification, so while the small number of tasks is a limitation of our work, we do analyze three times as many tasks as almost every LMC paper we cite. (The one arguable exception is [Wortsman et al.](https://arxiv.org/abs/2203.05482), which does test on text classification tasks but finds that their method does not work consistently outside of image classification.)
>
> As for the applications and implications: we believe that our results have significant applications for the currently popular practice of weight ensembling. They can also be applied to develop improved training methods by improving basin selection and factoring mechanistic generalization into the training process. In general, the machine learning community would benefit from an understanding of the loss surface that extends beyond the limited scope of image classification tasks.

---

> > ### Comment · Reviewer_bPGJ · 2022-12-12
> > **Response**
> >
> > Thanks for the response.
> >
> > While the question on whether models are trained identically or not has been properly addressed (and, in my opinion, the fact that they are trained identically makes some of the experimental observations even more surprising and interesting), I agree with some points raised by reviewer XZas, specifically on the main purpose of the paper and decisions on how different training phases are set up. Therefore, I am keeping my score.

---

> ### Author Response · Authors · 2022-12-07
> **Meta-review deadline**
>
> The AC has to write a meta-review by Sunday. We've addressed your questions on methodology and clarified them in the paper. Could I check if our answers are satisfactory? Thanks.

---

### Author Response · Authors · 2022-11-15
**general response**


We thank the reviewers for their time and their constructive comments. We hope our responses to their recommendations and troubles have improved the clarity of the paper.

- Multiple reviewers appear to be confused about how we generated the different models analyzed. Although we do describe this in the paper as it was submitted, we have further emphasized our explanation: the only difference between these models is that they have different random initialization seeds, which determine the different weights on the linear classifier head and the training data order. We do not deliberately introduce any generalization differences, and all differences come only from these different random seeds.
- Some of the reviewers also had trouble with the figure captions. We have added details to several captions to make it easier to follow the figures.

Other paper changes in response to specific comments are given in our replies.

---

### Decision · Program_Chairs · 2023-01-20

**Decision:**

Accept: poster

**Justification For Why Not Higher Score:**

This paper could easily be bumped up to spotlight, I am just being conservative due to the currently high score distribution.

**Justification For Why Not Lower Score:**

Reviewers are overall happy and scores are good, this is a clear accept, just I am not sure about spotlight vs poster.

**Metareview: Summary, Strengths And Weaknesses:**

This paper performs a detailed study of linear mode connectivity, primarily focusing on clusters of models which do satisfy linear connectivity.  The reviewers had some misgivings, but overall the paper was well-received, and I am happy to recommend clear acceptance.

**Note From Pc:**

if the above contains the word "oral" or "spotlight" please see: "oral" presentation means -> notable-top-5% and "spotlight" means -> notable-top-25%. As stated in our emails, we are disassociating presentation type from AC recommendations